# Generation of a bloodstream form *Trypanosoma brucei* double glycosyltransferase null mutant competent in receptor-mediated endocytosis of transferrin

**Samuel M. Duncan[1]¤, Carla Gilabert Carbajo[2], Rupa Nagar[1], Qi Zhong[2], Conor Breen[3], Michael A. J. Ferguson[1]\*, Calvin Tiengwe[2]\***

**1** Wellcome Centre for Anti-Infectives Research, School of Life Sciences, University of Dundee, Dundee, United Kingdom, **2** Faculty of Natural Sciences, Department of Life Sciences, Imperial College London, London, United Kingdom, **3** Regeneron Biotech, Raheen Business Park, Limerick, Ireland

¤ Current address: School of Biodiversity, One Health and Veterinary Medicine, University of Glasgow, Glasgow, United Kingdom
\* m.a.j.ferguson@dundee.ac.uk (MAJF); c.tiengwe@imperial.ac.uk (CT)

**Data Availability Statement:** The authors confirm that all data underlying the findings are fully available without restriction. All relevant data are

## Abstract

The bloodstream form of *Trypanosoma brucei* expresses large poly-*N*-acetyllactosamine (pNAL) chains on complex *N*-glycans of a subset of glycoproteins. It has been hypothesised that pNAL may be required for receptor-mediated endocytosis. African trypanosomes contain a unique family of glycosyltransferases, the GT67 family. Two of these, TbGT10 and TbGT8, have been shown to be involved in pNAL biosynthesis in bloodstream form *Trypanosoma brucei*, raising the possibility that deleting both enzymes simultaneously might abolish pNAL biosynthesis and provide clues to pNAL function and/or essentiality. In this paper, we describe the creation of a *TbGT10* null mutant containing a single *TbGT8* allele that can be excised upon the addition of rapamycin and, from that, a *TbGT10* and *TbGT8* double null mutant. These mutants were analysed by lectin blotting, glycopeptide methylation linkage analysis and flow cytometry. The data show that the mutants are defective, but not abrogated, in pNAL synthesis, suggesting that other GT67 family members can compensate to some degree for loss of TbGT10 and TbGT8. Despite there being residual pNAL synthesis in these mutants, certain glycoproteins appear to be particularly affected. These include the lysosomal CBP1B serine carboxypeptidase, cell surface ESAG2 and the ESAG6 subunit of the essential parasite transferrin receptor (TfR). The pNAL deficient TfR in the mutants continued to function normally with respect to protein stability, transferrin binding, receptor mediated endocytosis of transferrin and subcellular localisation. Further the pNAL deficient mutants were as viable as wild type parasites *in vitro* and in *in vivo* mouse infection experiments. Although we were able to reproduce the inhibition of transferrin uptake with high concentrations of pNAL structural analogues (*N*-acetylchito-oligosaccharides), this effect disappeared at lower concentrations that still inhibited tomato lectin uptake, i.e., at concentrations able to outcompete lectin-pNAL binding. Based on these findings, we recommend revision of the pNAL-dependent receptor mediated endocytosis hypothesis.

within the paper and its Supporting Information files.

**Funding:** This work was supported by a Wellcome Trust Investigator Award (101842/Z13/Z to MAJF) and by a Wellcome Trust and Royal Society Sir Henry Dale Fellowship (208780/Z/17/Z to CT, https://wellcome.org/). The funders had no role in the study design, data collection and analysis, decision to publish, or preparation of the manuscript.

**Competing interests:** The authors have declared that no competing interests exist.

## Author summary

Blood-stage trypanosome parasites have a specialised invagination on the cell surface named the flagellar pocket (FP), where invariant essential nutrient receptors are located. The pocket houses diverse proteins, including a transferrin receptor (TfR), which facilitates uptake of host transferrin-bound iron for survival. Several FP proteins, including TfR, are linked to complex sugar molecules (carbohydrates), the functions of which are not well understood. Complex carbohydrates are made by enzymes called glycosyltransferases (GTs) and previously we partially inhibited complex carbohydrate synthesis by deletion of either TbGT8 or TbGT10. However, mutant parasites lacking either one of these enzymes survived, suggesting functional redundancy. Here, we created a parasite mutant that lacks both TbGT8 and TbG10 to understand the combined effect of losing both enzymes. The mutant parasites showed a decreased ability to uptake tomato lectin, a protein that specifically binds to these sugar conjugates in the FP, indicating a reduction in carbohydrate complexity. Despite reduced complexity in the sugar structures attached to TfR, its critical function in transferrin/iron uptake remained effective. Furthermore, the mutants remained viable in culture and in animal models, challenging previous assumptions about the necessity and function of these carbohydrate conjugates. Our findings imply a greater flexibility and redundancy in the carbohydrate complex roles than previously appreciated.

## Introduction

The process of protein $N$-glycosylation in *T. brucei* has common and unique features compared to other eukaryotes, reviewed in [1] and summarised in (Fig 1). One unique feature is the synthesis of exceptionally large neutral poly-$N$-acetyllactosamine (pNAL) containing complex $N$-glycans by bloodstream form (BSF) trypomastigotes [2], the function(s) of which is/are unclear.

Protein $N$-linked glycosylation in eukaryotes requires the translocation of nascent proteins into the endoplasmic reticulum (ER), where Asn-Xaa-Ser/Thr sequons are $N$-glycosylated by an oligosaccharyltransferase (OST) which generally transfers $Glc_3Man_9GlcNAc_2$ from the lipid-linked oligosaccharide (LLO) $Glc_3Man_9GlcNAc_2$-PP-dolichol to the sidechain nitrogen of the Asn residue. Further, eukaryotic OSTs are generally hetero-oligomeric complexes composed of a catalytic STT3 subunit and up to eight additional subunits. However, in common with other trypanosomatid parasites [3,4], *T. brucei* OSTs only contain STT3 subunits. The *T. brucei* genome encodes three complete STT3 genes (*TbSTT3A*, *TbSTT3B* and *TbSTT3C*) and their protein products exhibit different specificities for their donor and acceptor substrates [5–7]. Thus, in BSF trypanosomes TbSTT3A acts first and transfers biantennary $Man_5GlcNAc_2$ from the LLO $Man_5GlcNAc_2$-PP-dolichol to sequons in acidic environments whilst TbSTT3B transfers triantennary $Man_9GlcNAc_2$ from LLO $Man_9GlcNAc_2$-PP-dolichol to remaining unoccupied sequons. The role of TbSTT3C is unclear as it is not well expressed in BSF or insect stage (procyclic) *T. brucei*, but from its expression in yeast it appears to have the LLO donor specificity of TbSTT3B with the acceptor sequon environment specificity of TbSTT3A [8,9].

Not only is protein $N$-glycosylation in *T. brucei* unusual in not utilising $Glc_3Man_9GlcNAc_2$-PP-dolichol as the OST LLO donor substrate, but the selective utilisation of $Man_5GlcNAc_2$-PP-dolichol and $Man_9GlcNAc_2$-PP-dolichol by TbSTT3A and TbSTT3B, respectively, means

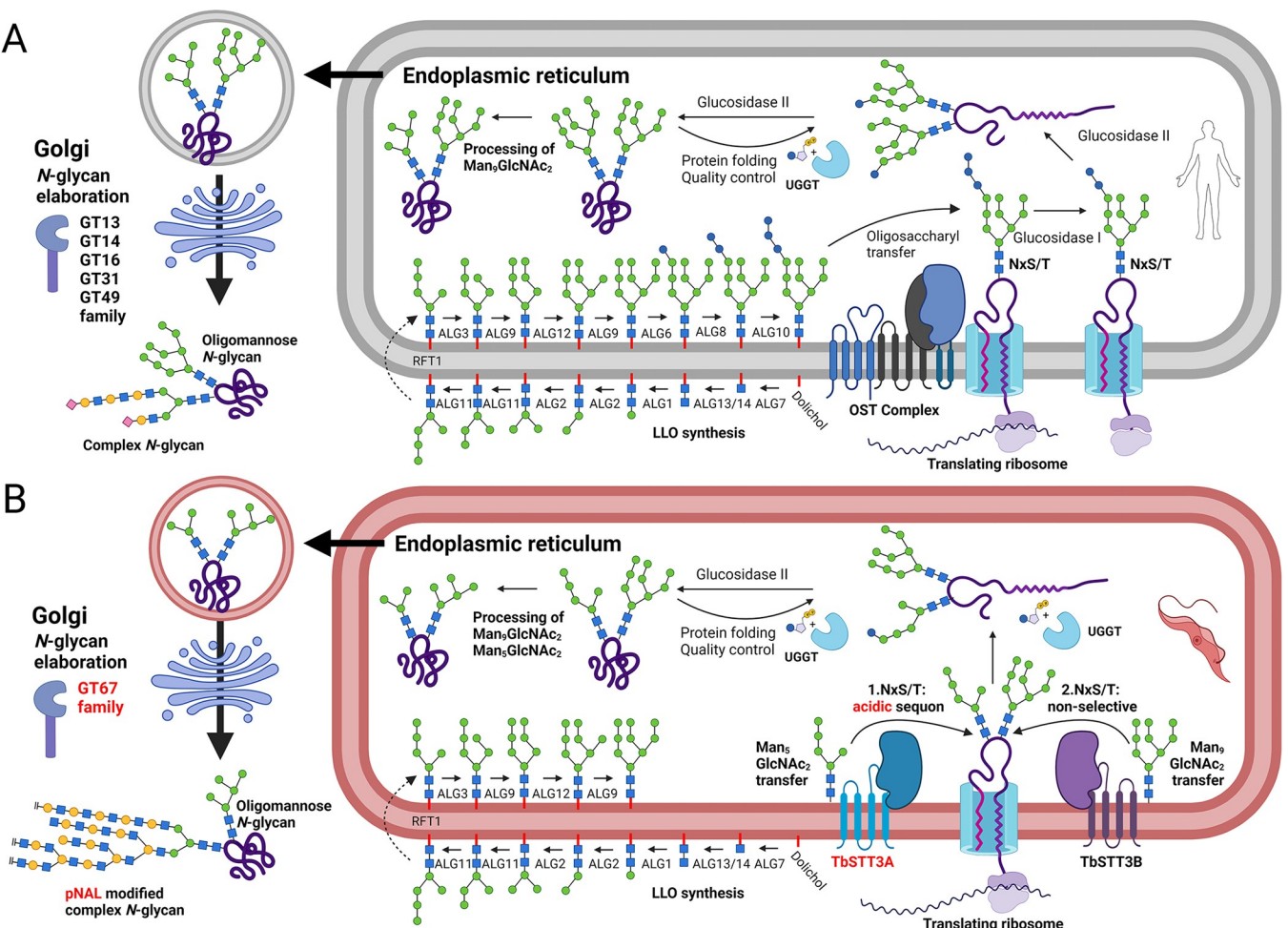

**Fig 1. Summary of the similarities and differences between general eukaryote protein *N*-glycosylation and processing systems (A) and those of bloodstream-form *T. brucei* (B).** Abbreviations: LLO, lipid-linked oligosaccharide; OST, oligosaccharyltransferase; pNAL, poly-N-acetyllactosamine; UGGT, UDP-Glc: glycoprotein glucosyltransferase. Adapted from [45], created with BioRender.com.

that this organism has evolved two simultaneously acting *N*-glycosylation pathways controlled by the amino acid environment of the *N*-glycosylation sequon (Fig 1).

Since *T. brucei* lacks an ER α-mannosidase II, the *N*-glycosylation sites occupied by triantennary $Man_9GlcNAc_2$ transferred via TbSTT3B can only be processed as far as triantennary $Man_5GlcNAc_2$ by ER α-mannosidase I. In other words, sites modified by TbSTT3B can only be occupied by triantennary $Man_{9-5}GlcNAc_2$ oligomannose glycans. On the other hand, sites occupied by biantennary $Man_5GlcNAc_2$ transferred via TbSTT3A can be processed to the paucimannose structure $Man_3GlcNAc_2$ by ER α-mannosidase I and, from there, to a variety of complex *N*-glycan structures up to and including large multi-antennary pNAL (Galβ1-4GlcNAc)$_n$ containing complex *N*-glycans with over 50 pNAL repeats [2] (Fig 1).

The elaboration of the *N*-linked paucimannose $Man_3GlcNAc_2$ structure into complex structures is performed by members of a large glycosyltransferase (GT) family located in the Golgi apparatus. This kinetoplastid-specific CAZy [10] GT67 family evolved from a single ancestral eukaryotic β3-GT gene to around 20 genes in *T. brucei* [1,11]. Despite their common origin, GT67 enzymes vary in their sugar nucleotide donor specificity (UDP-Gal or UDP-GlcNAc), their aglycone acceptors and the inter-sugar glycosidic linkages they catalyse (β1–2, β1–3, β1–

4 or β1–6). For example, TbGT8, TbGT3, TbGT11, TbGT15 and TbGT10 have been experimentally characterised [12–17] and, despite their common ancestry as GT67 family members, form the same glycosidic linkages as B3GNT1, B3GALT1, GlcNAc-transferase I (GnTI), GlcNAc-transferase II (GnTII), and GCNT2, enzymes that belong to different GT families (GT49, GT31, GT13, GT14, and GT16, respectively). By reverse genetics approaches we have deleted individual *TbGT* genes and characterised the *N*-glycan structures made by these mutants. The loss of certain TbGTs can lead to a compensatory elaboration of the truncated structure by the activity of the remaining TbGTs. For example, TbGT11 (TbGnTI) null mutants necessarily lack any elaboration on the 3-arm of the Manα1-3(Manα1–6)Manβ1-2GlcNAcβ1-4GlcNAc (Man$_3$GlcNAc$_2$) core structure, leading to increased branching activity and linear glycosylation on the 6-arm initiated by TbGT15 [14]. The opposite is observed in TbGT15 null mutants, where the absence of modification of the 6-arm results in increased 3-arm branching [16]. TbGT10 and TbGT8 function downstream of TbGT11 and TbGT15, alongside unidentified β4GalT(s), to synthesise GlcNAcβ1-6Gal [17] and GlcNAcβ1-3Gal [12] glycosidic linkages, respectively. Individually their activities extend linear pNAL (Galβ1-4GlcNAc) repeats, whilst their combined activity creates the GlcNAcβ1-6(GlcNAcβ1–3)Gal branch points. Individual deletion of *TbGT8* or *TbGT10* therefore impairs branching but elicits compensatory linear glycan synthesis via the remaining GT67 repertoire. Such compensatory extensions following specific TbGT deletion suggest that selective pressure is exerted on mutants to synthesise large pNAL containing *N*-glycans.

Why BSF trypanosomes have evolved to produce such unusually large and complex *N*-glycans is unclear. One proposal is that complex *N*-glycans containing pNAL repeats act as ligands for unspecified lectins that regulate in some way receptor-mediated endocytosis of scavenging receptors, such as the transferrin receptor (TfR) [18]. The process of scavenging iron from host transferrin (Tf) is essential to maintain cellular homeostasis and is achieved in BSF cells by expression of a cell surface TfR to scavenge holo-transferrin from the host as an iron source [19]. TfR has structural similarity to the major surface coat component of the parasite, the variant surface glycoprotein (VSG), but is a heterodimer of two proteins encoded by expression site associated genes *ESAG6* and *ESAG7* [20]. TfR is bound to the cell surface via a single glycosylphosphatidylinositol (GPI) anchor on ESAG6 [21,22] and localises to the flagellar pocket (FP), an invagination of the plasma membrane located at the base of the flagellum. TfR is heavily *N*-glycosylated; ESAG6 is occupied by two Man$_4$GlcNAc$_2$ paucimannose and three Man$_5$GlcNAc$_2$ oligomannose *N*-linked glycans, whilst ESAG7 is modified by a single paucimannose and two oligomannose structures [22]. Despite an earlier claim that TfR is devoid of pNAL containing *N*-glycans [22], more recent analysis demonstrated the reactivity of ESAG6 with lectins specific to pNAL modifications of complex *N*-glycans [23]. Nevertheless, evidence for the necessity of linear pNAL modifications of TfR as sorting signals for lectin-mediated endocytosis is limited. This hypothesis is based primarily on the reduced uptake of transferrin (Tf) in cells pre-treated with chito-oligosaccharides (polymers of (-4GlcNAcβ1-)$_n$) as surrogates of pNAL. Further, a counter argument for a key role for pNAL in the uptake of the essential nutrient Tf is that *T. brucei* BSF TbGT null mutants that affect the pNAL complex glycans [12,14,16,17] and BSF cells depleted of TbSTT3A by RNAi (lacking all or most complex *N*-glycans [5]) have no significant growth phenotypes in culture and the TbGT null mutants can all efficiently infect mice. Given this discrepancy, we considered that a more targeted mutational approach might resolve the issue.

To this end we generated a *TbGT10*$^{-/-}$/*TbGT8*$^{Flox/-}$ conditional null mutant by diCre/LoxP mediated recombination in a *TbGT10*$^{-/-}$ null mutant background [17]. From this, we also isolated a *TbGT10*$^{-/-}$/*TbGT8*$^{-/-}$ double null mutant. Based on previous structural characterisation of *N*-glycans from the *TbGT8*$^{-/-}$ null [12,13] and *TbGT10*$^{-/-}$ null [17] mutants, double deletion of *TbGT8*

and *TbGT10* should, theoretically, prevent the synthesis of both linear pNAL chains and of 3,6 branch-points. Here we describe the phenotype of BSF *T. brucei* simultaneously lacking *TbGT8* and *TbGT10* and discuss the implications for the role of pNAL synthesis in this organism.

## Results

### Generation of a bloodstream form TbGT8 conditional null mutant in a TbGT10 null mutant background

The generation of a *TbGT10* null mutant ($\Delta gt10$::*PAC*/$\Delta gt10$ (ribosomal small subunit (*SSU*) *diCre*)) in a cell line expressing diCre recombinase has been previously described [17] (Fig 2A). Starting with this parental cell line, the first *TbGT8* allele was replaced by homologous recombination using a blasticidin deaminase (*BSDr*) resistance cassette flanked by tubulin regulatory elements to generate a *TbGT8*$^{+/-}$ heterozygote. The remaining allele was replaced with a construct containing a *TbGT8* gene flanked by loxP sites (floxed) in tandem with a hygromycin phosphotransferase-thymidylate kinase (*HYG-TK*) cassette for negative and positive selection, respectively [24]. A $\Delta gt10$::*PAC*/$\Delta gt10$/$\Delta gt8$::*BSDr*/$\Delta gt8$::*TbGT8-HYG-TK*$^{Flox}$ [*SSU diCre*] clone was selected (hereafter referred to as *TbGT10*$^{-/-}$/*TbGT8*$^{Flox/-}$) expressing a single floxed copy of *TbGT8*. This cell line is a *TbGT8* conditional null mutant in a *TbGT10* null background, where *TbGT8* can be excised upon addition of rapamycin. This was demonstrated by genomic DNA extraction and PCR analysis (Fig 2B). The data show the *TbGT8* locus in wild type cells, the *TbGT10* KO$^{-/-}$/*TbGT8*$^{+/-}$ mutant and two independent *TbGT10*$^{-/-}$/*TbGT8*$^{Flox/-}$ conditional null mutant clones grown in the absence and presence of rapamycin for three days. The rapamycin-induced excision of the *TbGT8*$^{Flox}$ allele in both clones is apparent (Fig 2B, 1.1 and 1.2). We also assessed the expected loss of hygromycin (HYG) resistance following rapamycin treatment. The majority of rapamycin-treated cells were killed by hygromycin, confirming high gene excision efficiency (Fig 2C). The *TbGT10*$^{-/-}$/*TbGT8*$^{Flox/-}$ conditional null mutant showed similar growth kinetics in the absence and presence of rapamycin (Fig 2D). This indicates that, under *in vitro* culture conditions, BSF *T. brucei* can survive without *TbGT8* and *TbGT10* individually [12,17] and collectively (Fig 2D).

### A stable TbGT10 and TbGT8 double null mutant is viable in mice

A stable *TbGT10* and *TbGT8* double null mutant cell line was generated by treating *TbGT10*$^{-/-}$/*TbGT8*$^{Flox/-}$ clone 1.1 cells with 100 nM rapamycin in the absence of hygromycin for 3 days and selecting clones by limiting dilution. The absence of both *TbGT10* and *TbGT8* was confirmed by hygromycin sensitivity and PCR amplification of their respective loci (Fig 3A), alongside a control *TbGT2b* amplicon. A *TbGT10*$^{-/-}$/*TbGT8*$^{-/-}$ double null mutant clone was selected for further analysis. The absence of both *TbGT10* and *TbGT8* was confirmed in this mutant by whole genome sequencing analysis (S1 Fig).

We analysed the infectivity of the *TbGT10*$^{-/-}$/*TbGT8*$^{-/-}$ double null mutant in mice. No difference in the ability of wild type or *TbGT10*$^{-/-}$/*TbGT8*$^{-/-}$ double null mutant cells to infect Balb/c mice was observed (Fig 3B). These data demonstrate that neither the individual [12,17] nor collective activities of TbGT8 and TbGT10 (Fig 3B) are essential for the survival of BSF *T. brucei in vivo*.

### Excision of TbGT8 in a TbGT10 null background alters protein glycosylation

Based on the glycosidic linkages catalysed by TbGT10 and TbGT8 [12,17], and previous structural studies on *N*-glycans from BSF *T. brucei* [1], we reasoned that cells simultaneously

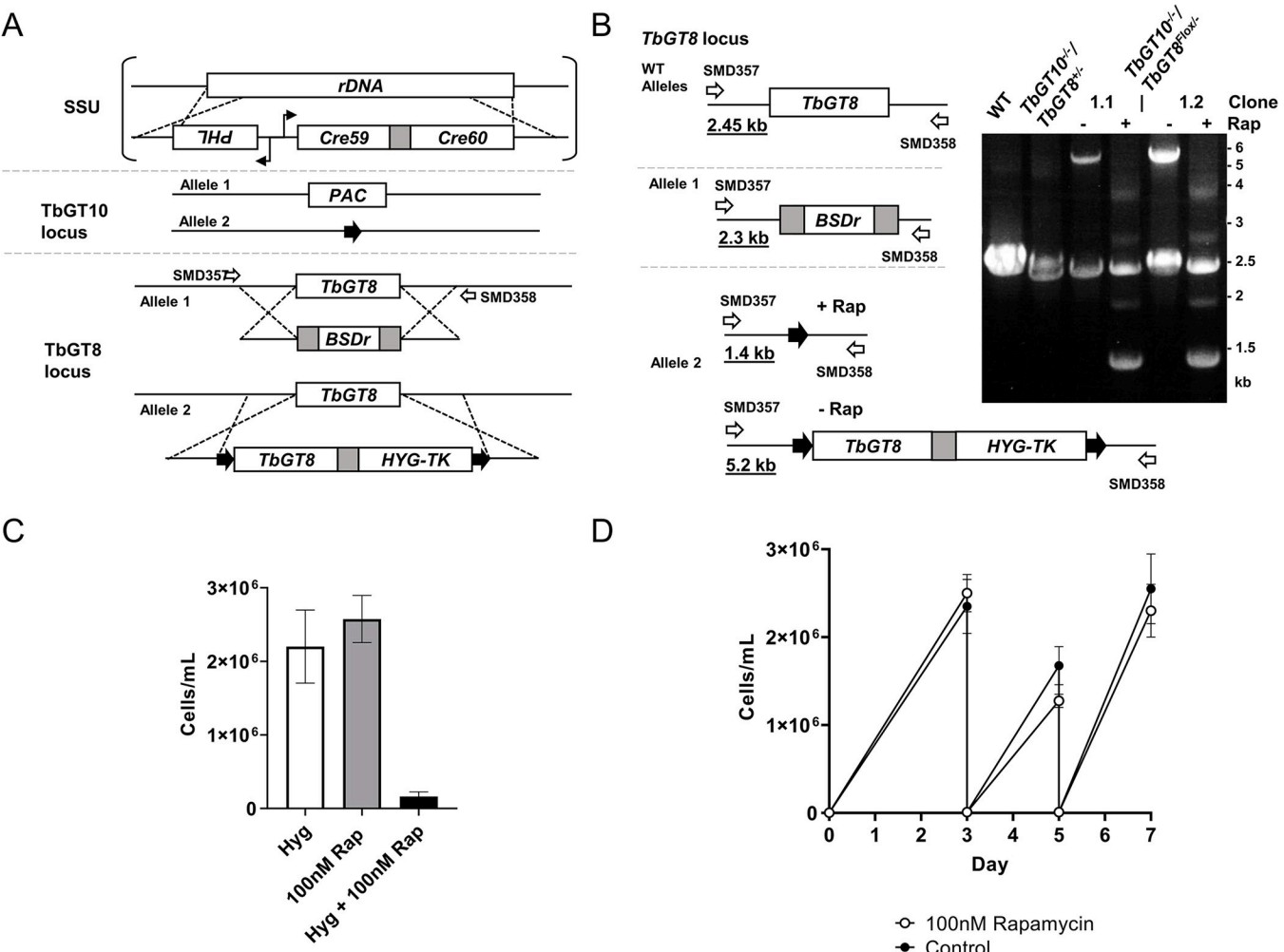

**Fig 2. Generation of a bloodstream form *TbGT8* conditional null mutant in a *TbGT10* null mutant background. A.** Gene replacement strategy using a previously described *TbGT10* null cell line (*TbGT10^-/-^*) [17] which constitutively expresses diCre recombinase from the ribosomal small subunit (SSU). This was used to generate a TbGT8 conditional knockout cell line lacking *TbGT10* (referred to in the text as *TbGT10^-/-^/TbGT8^Flox/-^* conditional null). The first allele of *TbGT8* was replaced using a blasticidin resistance gene (*BSDr*) to generate *TbGT10^-/-^/TbGT8^+/-^* mutants. To make a conditional null mutant of *TbGT8* the transgene was introduced at the second allele using a dicistronic construct containing a hygromycin resistance (*HYG*) Thymidine kinase (*TK*) fusion flanked by *loxP* (black arrows) sites (*TbGT8^Flox^*). **B.** PCR was performed using gDNA harvested 72 h after rapamycin treatment (+ Rap) and oligonucleotide primers SMD357 and 358 (open arrows) that anneal outside of the homologous recombination site. Expected amplicon sizes are underlined. Resolution of PCR products by agarose gel electrophoresis confirms the replacement of endogenous *TbGT8* by *TbGT8^Flox^* and the excision of *TbGT8^Flox^* upon rapamycin treatment. Two *TbGT10^-/-^/TbGT8^Flox/-^* clones (1.1 and 1.2) are shown. Wild-type (WT) and *TbGT10^-/-^/TbGT8^+/-^* mutants were included as controls. **C.** Growth of *TbGT10^-/-^/TbGT8^Flox/-^* conditional null mutant cells cultured with (+Rap) or without (−Rap) 100 nM rapamycin for 3 days. Cells were seeded in the presence or absence of hygromycin (HYG) to assess floxed gene loss by hygromycin sensitivity. Data are means ± SD (n = 4 clones with 3 technical replicates per n) **D.** Growth kinetics of *TbGT10^-/-^/TbGT8^Flox/-^* conditional null mutant cells grown with or without (control) 100 nM rapamycin for 7 days. Cells were seeded at $2 \times 10^3$ cells/ml and diluted to $2 \times 10^4$ cells/ml every 2 days from day 3. Cell density was determined by counting at 24 h intervals. Data are means ± SD (n = 4 clones with 3 technical replicates each).

lacking TbGT10 and TbGT8 might be restricted to synthesising of tri-antennary oligomannose ($Man_{5-9}GlcNAc_2$) structures, bi-antennary paucimannose structures and small bi-antennary ($Gal\beta1\text{-}4GlcNAc\beta1–2)_2Man_3GlcNAc_2$ complex *N*-glycans devoid of pNAL (Fig 4A), some of which might be capped on either or both arms with an αGal residue in 1–3 linkage [25].

We performed lectin blotting with ricin agglutinin (RCA) on SDS cell lysates of wild type cells and of two independent *TbGT10^-/-^/TbGT8^Flox/-^* conditional null mutant clones grown without or with rapamycin for 3 days. We have previously shown that deletion of TbGT10

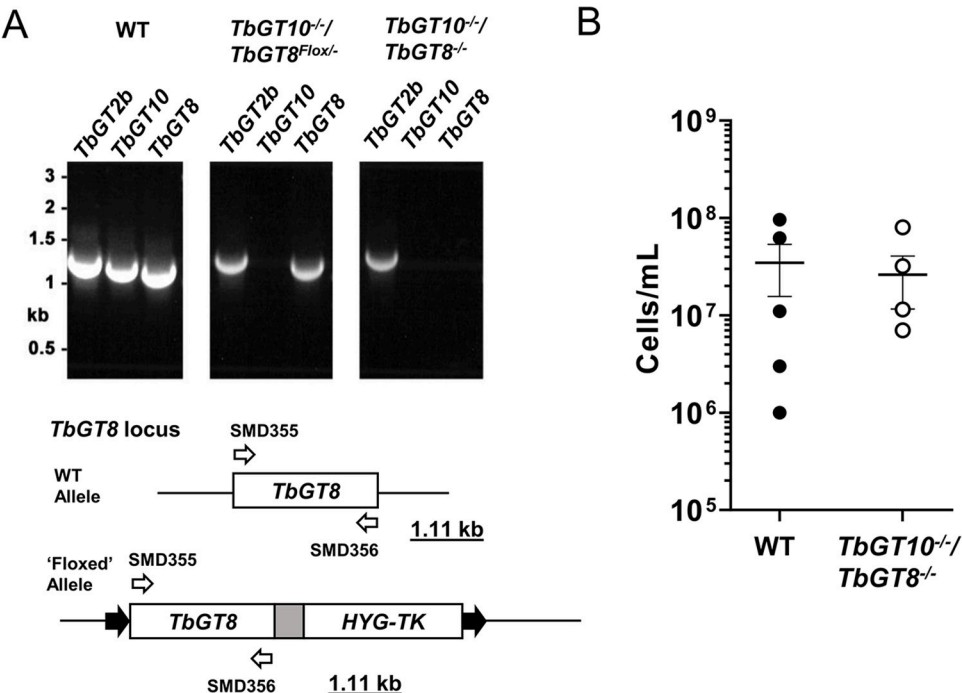

**Fig 3. A stable *TbGT10/TbGT8* double KO clone is viable in mice. A.** PCR was performed using gDNA extracted from wild-type (WT), *TbGT10*[-/-]/*TbGT8*[Flox/-] conditional null mutant and *TbGT10*[-/-]/*TbGT8*[-/-] double null mutant (clone 1.1) cells and oligonucleotide primers SMD357 and 358 (open arrows) that amplify *TbGT8*. Expected amplicon sizes are underlined. PCR amplification of *TbGT2b* and *TbGT10* coding sequences was carried out in parallel as controls. Resolution of PCR products by agarose gel electrophoresis confirms the loss of both *TbGT10* and *TbGT8* from the *TbGT10*[-/-]/*TbGT8*[-/-] double null mutant. **B.** Balb/c mice were infected with 2 x10^5 WT and *TbGT10*[-/-]/ *TbGT8*[-/-] double null mutant cells and viable cells were counted 3 days post infection. No significant difference was observed between WT and *TbGT10*[-/-]/*TbGT8*[-/-] double null mutant cells. Data are means ± SD (n = 4–5 mice with 3 technical replicates per n).

results in a reduction in RCA binding (implying a decrease in terminal βGal residues) and an alteration in the banding pattern of glycoproteins above ~70 kDa compared to wild-type [17]. We observed the same alteration in RCA binding pattern in the lysates of both *TbGT10*[-/-]/ *TbGT8*[Flox/-] conditional null mutant clones grown under permissive (- rapamycin) conditions compared to wild type (Fig 4B, compare lanes 2 and 4 with lane 1). The unaffected band at ~55 kDa band is most likely VSG221, which carries a terminal βGal residue on its GPI anchor [26]. The most striking additional feature upon rapamycin-induced excision of *TbGT8* in the *TbGT10* null background was the appearance of a new RCA-binding band at ~60 kDa indicated by an asterisk (Fig 4B, lanes 3 and 5). The same effect was seen when comparing lysates from wild type, a *TbGT10*[-/-]/*TbGT8*[Flox/-] conditional null mutant grown under permissive conditions and the *TbGT10*[-/-]/*TbGT8*[-/-] double null mutant cells (Fig 4C).

Bloodstream form parasites express surface glycoproteins such as the ESAG6/7 heterodimeric transferrin receptor (TfR) and ISG65 that resolve by SDS-PAGE in the region of the RCA-binding ~60 kDa glycoprotein(s) from *TbGT10*[-/-]/*TbGT8*[-/-] cells. We therefore checked the status of these glycoproteins in lysates of wild type, a *TbGT10*[-/-]/*TbGT8*[Flox/-] conditional null mutant grown under permissive conditions and the *TbGT10*[-/-]/*TbGT8*[-/-] double null mutant cells (Fig 4C) by anti-ISG65 and anti-TfR Western blotting (S2 Fig). This revealed that, in all cases, ISG65 runs at a distinctly higher apparent molecular weight than the RCA-binding ~60 kDa glycoprotein(s) from *TbGT10*[-/-]/*TbGT8*[-/-] cells. Further, it showed that ISG65 is not

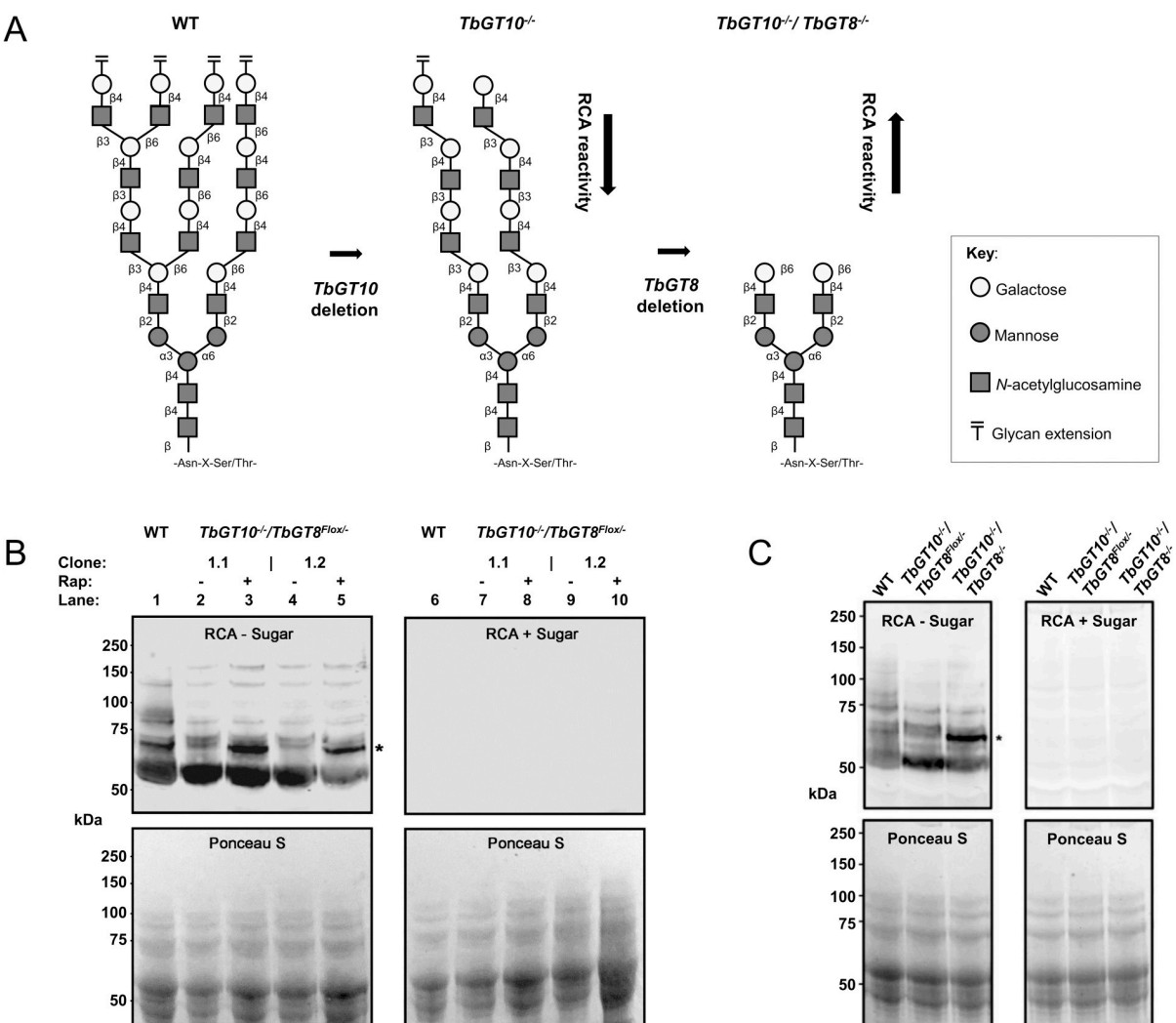

**Fig 4. Reactivity of *TbGT* mutant cell lysates with RCA lectin. A.** Schematic shows the predicted outcome of double gene deletion *TbGT10* and *TbGT8* on BSF complex *N*-glycan elaboration based on previous analysis of their glycosyltransferase activity [8,13]. GlcNAc: βGal transferase activities due to TbGT8 and TbGT10 deletion prevents glycolinkages 16 and 17, respectively, resulting in deficiencies in the synthesis of 4GlcNAcβ1-3(-4GlcNAcβ1–6)Galβ1-branch points in *N*-linked glycans. Thus, loss of either glycosyltransferase activity similarly prevents branch point formation, and loss of both is predicted to inhibit the synthesis of linear pNAL, leading to the synthesis of *N*-glycans terminating in RCA-reactive β4-Galactose. **B.** Lysates of WT, *TbGT10⁻/⁻/TbGT8^Flox/-* conditional null mutant and *TbGT10⁻/⁻/TbGT8⁻/⁻* double null mutants were subjected to SDS-PAGE and transferred to nitrocellulose membrane in duplicate. *Upper panel*, membranes were incubated with biotinylated RCA without (RCA—sugar) or with pre-incubation with 30 mg/ml galactose and lactose (RCA + sugar) as a binding specificity control. *Lower panels*, equal loading and transfer are demonstrated by Ponceau S staining. The molecular weight markers are indicated on the left. Increased detection of a discrete ~60 kDa product by RCA was observed in *TbGT10⁻/⁻/TbGT8⁻/⁻* double null mutant lysates (marked by *). **C.** Lysates of WT, *TbGT10⁻/⁻/TbGT8^Flox/-* conditional null and *TbGT10⁻/⁻/TbGT8⁻/⁻* double null mutants were subjected to SDS-PAGE and transferred to nitrocellulose membrane in duplicate. *Upper panel*, membranes were incubated with biotinylated RCA without (RCA—sugar) or with pre-incubation with 30 mg/ml galactose and lactose (RCA + sugar) as a binding specificity control. *Lower panels*, equal loading and transfer are demonstrated by Ponceau S staining. The molecular weight markers are indicated on the left. An enhanced signal for a distinct ~60 kDa species (marked by *) in *TbGT10⁻/⁻/TbGT8⁻/⁻* double null mutant cell lysates was again observed.

perturbed by the absence of TbGT10 or the absence of TbGT10 and TbGT8. This suggests that ISG65 lacks elaborated complex *N*-glycans. On the other hand, the anti-TfR Western blot showed that the absence of TbGT10 reduces the apparent molecular weight, and heterogeneity, of the ESAG6 subunit and that this is even more marked in the *TbGT10⁻/⁻/TbGT8⁻/⁻* double null cell line. The effects on TfR of TbGT10 were not explored in our previous TbGT10 study

[17]. However, in all cases, the TfR subunits did not comigrate with the RCA-binding ~60 kDa glycoprotein(s) from *TbGT10*$^{-/-}$/*TbGT8*$^{-/-}$ cells.

To try to identify the RCA-binding ~60 kDa glycoprotein(s) from *TbGT10*$^{-/-}$/*TbGT8*$^{-/-}$ cells, we performed hypotonic lysis of wild type and *TbGT10*$^{-/-}$/*TbGT8*$^{-/-}$ cells to separate sVSG-enriched soluble fractions and cell ghost pellets. Proteins from each fraction were resolved by SDS-PAGE and subjected to RCA lectin blotting. The RCA-binding ~60 kDa glycoproteins enriched in the *TbGT10*$^{-/-}$/*TbGT8*$^{-/-}$ cells were found predominantly in the supernatant fraction (S3A Fig), suggesting that they are either GPI-anchored, and thus released in a soluble form upon hypotonic cell lysis (like sVSG), and/or lumenal soluble glycoproteins in the secretory or endosomal membrane system and released upon hypotonic organelle rupture. To identify them, we performed RCA-agarose pulldowns from the supernatant fractions of wild type, a *TbGT10*$^{-/-}$/*TbGT8*$^{Flox/-}$ conditional null mutant grown under permissive conditions and *TbGT10*$^{-/-}$/*TbGT8*$^{-/-}$ double null mutant cells. A preclearance step using anti-VSG IgG conjugated to Protein G agarose was performed to remove the majority of sVSG. The remaining RCA-binding glycoproteins were resolved by SDS-PAGE in triplicate for Coomassie staining and RCA lectin blotting with and without inhibitory sugars. The RCA reactive products specifically enriched in the *TbGT10*$^{-/-}$/*TbGT8*$^{-/-}$ sample were excised from the gel, along with the equivalent sections from the wild type and *TbGT10*$^{-/-}$/*TbGT8*$^{Flox/-}$ conditional null mutant grown under permissive conditions (S3B Fig) and processed for proteomic analysis by mass spectrometry. Protein groups enriched in the *TbGT10*$^{-/-}$/*TbGT8*$^{-/-}$ sample, relative to the wild type and *TbGT10*$^{-/-}$/*TbGT8*$^{Flox/-}$ conditional null mutant grown under permissive conditions samples, are indicated in (Table 1). Among these, two (ESAG2 and CBP1B) had predicted *N*-terminal signal peptides and 6 and 7 predicted *N*-glycosylation sites, respectively [27] (S4 Fig),

**Table 1. List of identified proteins from the RCA reactive band near 60 kDa.** emPAI values scores are indicated for WT, *TbGT10*$^{-/-}$/*TbGT8*$^{Flox/-}$ conditional null and *TbGT10*$^{-/-}$/*TbGT8*$^{-/-}$ double null mutant cells. Non-detected (nd) proteins are indicated. Prediction of a Signal Peptide (SP) by SignalP analysis indicated. *N*-glycosylation site prediction was performed [27]. Expanded sequence analysis of ESAG2 and CBP1B (bold) are shown in S4 Fig.

| Gene ID | Mass (da) | Product Description | SP | WT | *TbGT10*$^{-/-}$/ *TbGT8*$^{Flox/-}$ | *TbGT10*$^{-/-}$ / *TbGT8*$^{-/-}$ | *N*-linked glycans |
|---|---|---|---|---|---|---|---|
| Tb927.11.2610 | 51038 | Cytoskeleton-associated protein 50 kDa | N | 78.1 | 61.25 | 92.03 | |
| Tb927.7.2640 | 51052 | Cytoskeleton associated protein 51 | N | 6.56 | 6.58 | 9.46 | |
| **Tb927.10.1040** | **52276** | **serine peptidase, Clan SC, Family S10 (CBP1B)** | **Y** | **1.79** | **2.84** | **6.81** | **7** |
| Tb927.8.7970 | 53692 | hypothetical protein | N | 0.26 | 0.71 | 1.33 | |
| **Tb927.11.14620** | **56086** | **expression site-associated gene 2 (ESAG2) protein, putative** | **Y** | **nd** | **0.81** | **1.81** | **6** |
| Tb927.11.1980 | 58937 | zinc finger protein family member, putative | N | 15.6 | 6.15 | 18.06 | |
| Tb927.10.1060 | 59031 | T-complex protein 1, delta subunit, putative | N | 7.77 | 3.06 | 5.18 | |
| Tb927.11.3240 | 59877 | T-complex protein 1, zeta subunit, putative | N | 38 | 11.08 | 43.84 | |
| Tb927.7.2650 | 62172 | Cytoskeleton associated protein 51V | N | 23.4 | 21.9 | 43.46 | |
| Tb927.9.10770 | 62423 | polyadenylate-binding protein 2 | N | 709 | 144.3 | 1799.5 | |
| Tb927.11.9210 | 63191 | NOL1/NOP2/sun family, putative | N | 0.3 | 0.69 | 1.34 | |
| Tb927.9.9290 | 63499 | polyadenylate-binding protein 1 | N | 32.8 | 23.44 | 52.31 | |
| Tb927.5.1250 | 64339 | GAF domain/TIP41-like family, putative | N | 0.57 | 0.78 | 1.62 | |
| Tb927.11.11700 | 68560 | nucleotidyl transferase, putative | N | 1.79 | 1.19 | 3.01 | |
| Tb927.10.6320 | 68681 | CBF/Mak21 family, putative | N | 0.94 | 0.83 | 2.34 | |
| Tb927.6.4590 | 69413 | glutamyl-tRNA synthetase, putative | N | 1.05 | 1.45 | 2.11 | |
| Tb927.5.4380 | 71995 | Kinetoplastid-specific Protein Phosphatase 1 | N | 0.68 | 0.33 | 1.66 | |
| Tb427.BES40.9 | 77685 | expression site-associated gene 8 (ESAG8) protein, putative | N | nd | nd | 7 | |

indicative of Golgi trafficking and elaboration by glycosyltransferases. Both are known to resolve at high molecular weight by SDS-PAGE in wild type BSF cell lysates [13,18].

From these data, we conclude that at least some glycoproteins are affected as predicted in (Fig 4A), collapsing from very high apparent MW to, in the case of ESAG2 and CBP1B, around 60 kDa apparent MW upon loss of both TbGT10 and TbGT8. Consistent with their appearance in the soluble fraction upon hypotonic lysis, ESAG2 is predicted to be a GPI-anchored protein [27] while CBP1B is a soluble lysosomal/endosomal serine protease [28].

## The roles of TbGT10 and TbGT8 in poly-N-acetyllactosamine (pNAL) expression

A characteristic of BSF *T. brucei* is the presence of very large pNAL containing *N*-linked glycans on glycoproteins of the flagellar pocket and endosomal and lysosomal compartments [1,2]. To address the impact of removing TbGT10 and TbGT8 on pNAL expression, we performed tomato lectin (TL) blots of whole cell lysates, TL flow cytometry and methylation linkage analysis on glycopeptide preparations.

Blotting of whole cell lysates with biotinylated TL showed that deletion of TbGT10 (using the *TbGT10<sup>-/-</sup>/TbGT8<sup>Flox/-</sup>* conditional null mutant grown under permissive conditions) resulted in a significant reduction in the overall TL signal intensity for glycoproteins larger than 50 kDa (Fig 5A, lane 2) compared to wild type (Fig 5A, lane 1). The reduction in TL binding is incomplete, however, with chitin hydrolysate-inhibitable signals remaining at around 75, 100, 150 and >250 kDa apparent molecular weight (Fig 5A, lane 2). The further ablation of TbGT8 in this TbGT10 null background had little discernible effect on the TL blot pattern (Fig 5A, lane 3). The signals below 50 kDa correspond to endogenously biotinylated proteins and are not inhibited by chitin hydrolysate.

Next, we quantitatively assessed the endocytic capacity of these parasites using flow cytometry to measure binding and uptake of fluorescent-conjugated TL as a surrogate marker for receptor-mediated endocytosis (Fig 5B and 5C). Cells were incubated with DyLight 488-tomato lectin (TL::488) at 4˚C to assess binding at the flagellar pocket without internalisation [29]. The cells were subsequently transferred to either 14˚C or 37˚C for 10 minutes to activate endocytosis, directing TL::488 either to intermediate endocytic compartments or the terminal lysosome, respectively [29]. At 4˚C, TL binding remained unchanged in *TbGT10<sup>-/-</sup>/TbGT8<sup>Flox/-</sup>* conditional null mutant cells grown under permissive conditions but was reduced by 42% in *TbGT10<sup>-/-</sup>/TbGT8<sup>-/-</sup>* cells ($p < 0.01$), compared to wild type cells (Fig 5B and 5C). At 14˚C and 37˚C, TL uptake was significantly impaired in both TbGT mutants relative to wild type cells. In *TbGT10<sup>-/-</sup>/TbGT8<sup>Flox/-</sup>* conditional null mutant cells grown under permissive conditions, the reductions were 52% at 14˚C, $p = 0.0029$ and 64%, $p = 0.0078$ at 37˚C, while in *TbGT10<sup>-/-</sup>/TbGT8<sup>-/-</sup>* cells, the reductions were 64%, $p = 0.0033$ at 14˚C and 79%, $p = 0.0046$ at 37˚C (Fig 5C). To further confirm the reductions in TL:488 binding and uptake, WT and mutant cells were assessed for their capacity to efficiently endocytose by fluorescence microscopy. The results showed reduced TL signal intensity adjacent to the kinetoplast (flagellar pocket) in the mutants relative to WT cells at 4˚C, with minimal or undetectable signal in endocytic compartments (between the nucleus and the kinetoplast) at 37˚C when using wild-type TL::488 fluorescence detection limits (S5 Fig). The significant decrease in TL binding in the *TbGT10<sup>-/-</sup>/TbGT8<sup>-/-</sup>* cells at all temperatures, compared to *TbGT10<sup>-/-</sup>/TbGT8<sup>Flox/-</sup>* conditional null mutant cells grown under permissive conditions and wild type cells, suggests that TbGT8 has a more profound effect on TL-reactive pNAL expression in a cellular context than is discernible by lectin blotting.

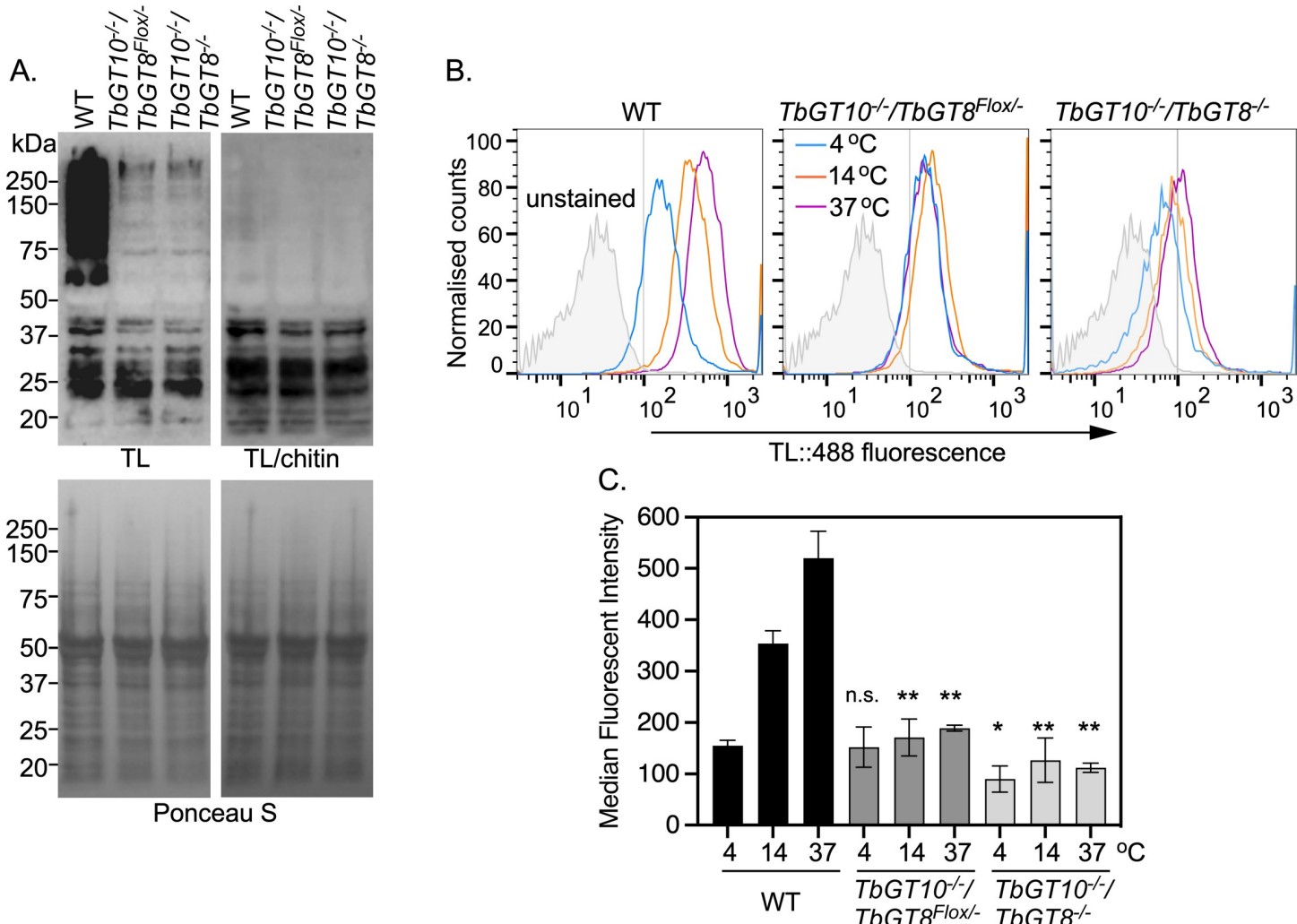

**Fig 5. TbGT knockout mutants are deficient in TL binding and uptake. A.** Whole cell lysates ($10^7$ cell equivalents per condition) from WT, *TbGT10⁻/⁻/TbGT8^Flox/-* conditional null mutant and *TbGT10⁻/⁻/TbGT8⁻/⁻* double null mutant cell lines were subjected to SDS-PAGE, transferred to PVDF membranes, and probed with TL::Biotin only (TL, top left) or with TL::Biotin pre-incubated with chitin hydrolysate at 1:10 dilution (TL/chitin, top right) as competitive inhibitor. Membranes were counterstained with Ponceau S as loading control (bottom panels). **B.** Binding and internalisation of TL::Dylight488, acting as a surrogate for receptor-mediated endocytic cargoes, were measured by flow cytometry. Cells were incubated with TL::Dylight488 at 4°C for 5 min and then transferred to 14 or 37°C for 10 min to activate endocytosis. Histogram profiles show fluorescent intensities distributions of 10,000 WT, *TbGT10⁻/⁻/TbGT8^Flox/-* conditional null mutant and *TbGT10⁻/⁻/TbGT8⁻/⁻* double null mutant cells. **C.** Bar chart shows median fluorescence intensities (MFI, arbitrary units) for each population of cells, presented as means ± SD, n = 3 biological replicates. Unpaired t-test with Welch's correction indicated significant differences (*$P < 0.0339$, **$P < 0.008$) between WT and each of the mutant cell lines at the specified temperatures. Chitin hydrolysate at 1:10, 1:100 and 1:000 dilution (inhibitor) was used as specificity control for each temperature and the MFI values were < 22 arbitrary units.

The TL dependent parameters measured above are potentially complicated by the specificity of tomato lectin, which has high-affinity for linear pNAL structures of 3 or more [-3Galβ1-4GlcNAcβ1-] repeats [30] but which also binds to the *N*-acetyl-chitobiose core of oligomannose of *N*-linked glycans in lectin blots [31]. To address this, we investigated the *N*-glycan structures synthesised by the *TbGT10⁻/⁻/TbGT8⁻/⁻* double null mutant compared to wild type using a chemical method. Following hypotonic lysis, to remove the majority of VSG as sVSG, washed cell ghost *N*-glycopeptide fractions were prepared by Pronase digestion, as previously described in [17] and subjected to methylation linkage analysis by gas chromatography-mass spectrometry (GC-MS). The partially permethylated alditol acetate (PMAA) derivatives were

**Table 2. GC-MS methylation linkage analysis of glycopeptides from BSF wild type (WT) and *TbGT8⁻ᐟ⁻/TbGT10⁻ᐟ⁻* double null (dKO) mutants.** The PMAAs derivatives measured and the respective characteristic ions used for analysis in Fig 6.

| PMAA derivatives | Residue types | Characteristic ion used for analysis; WT (Heavy)/double null mutant (Light) |
|---|---|---|
| [1-²H]-1,5-Di-*O*-acetyl-2,3,4,6-tetra-*O*-methylmannitol | t-Man | *m/z* 145/148 |
| [1-2H]-1,5-Di-*O*-acetyl-2,3,4,6-tetra-*O*-methylgalactitol | t-Gal | *m/z* 145/148 |
| [1-2H]-1,3,5-Tri-*O*-acetyl-2,4,6-tri-*O*-methylgalactitol | 3-Gal | *m/z* 161/163 |
| [1-2H]-1,5,6-Tri-*O*-acetyl-2,3,4-tri-*O*-methylgalactitol | 6-Gal | *m/z* 162/164 |
| [1-²H]-1,3,5,6-Tetra-*O*-acetyl-2,4-di-*O*-methylmannitol | 3,6-Man | *m/z* 234/236 |
| [1-2H]-1,3,5,6-Tetra-*O*-acetyl-2,4-di-*O*-methylgalactitol | 3,6-Gal | *m/z* 234/236 |
| [1-2H]-1,4,5-Tri-*O*-acetyl-2-methylacetamido-3,6-di-*O*-methylglucosaminitol | 4-GlcNAc | *m/z* 159/160 |

normalised to the PMAA corresponding to non-reducing-terminal mannose (t-Man) arising from oligomannose structures (Table 2 and Fig 6). As expected, the relative 3,6-disubstituted Man (3,6-Man) derivative levels are similar, since 3,6-Man is present in both oligomannose and complex *N*-glycan structures. In contrast, there is a significant reduction in the relative levels of the Gal and GlcNAc PMAA derivatives, consistent with a reduction in *N*-glycan processing to large complex structures. Specifically, there is a reduction in t-Gal, 3-Gal, 6-Gal, 3,6-Gal and 4-GlcNAc residues consistent with a reduction in the UDP-GlcNAc: βGal β1–6 and β1–3 transferase activities of TbGT10 and TbGT8, respectively. However, the residual levels of 3-Gal, 6-Gal, 3,6-Gal residues make it clear that other TbGTs, most likely encoded within the Carbohydrate Active enZymes (CAZy) GT67 family [1], can substitute to some degree for TbGT10 and TbGT8 and that the *TbGT10⁻ᐟ⁻/TbGT8⁻ᐟ⁻* double null mutant is, therefore, not devoid of pNAL containing *N*-linked glycans (S6 and S7 Figs).

## pNAL expression on the BSF transferrin receptor is greatly reduced in the TbGT10⁻ᐟ⁻ and TbGT10⁻ᐟ⁻/TbGT8⁻ᐟ⁻ null mutants but transferrin-binding, localisation and receptor-mediated endocytosis are not affected

The identification of CBP1B and ESAG2 in the RCA-binding 60 kDa apparent molecular weight material found in the *TbGT10⁻ᐟ⁻/TbGT8⁻ᐟ⁻* double null mutant (Fig 4B and 4C) suggested that some glycoproteins might be more affected than others with respect to pNAL content in the absence of TbGT10 and TbGT8. Further, a proposed role for pNAL modified *N*-glycans of BSF parasites is as sorting signal for lectin-mediated endocytosis of the TfR [18]. We therefore compared the glycosylation status, TfR binding, localisation and receptor mediated endocytic properties of TfR in wild type, *TbGT10⁻ᐟ⁻/TbGT8^{Flox/-}* conditional null mutant grown under permissive conditions and *TbGT10⁻ᐟ⁻/TbGT8⁻ᐟ⁻* double null mutant cells.

While newly synthesized ESAG6 and ESAG7 show weak TL reactivity via their oligomannose and paucimannose *N*-glycans, only *N*-glycans of ESAG6 are extensively elaborated by addition of pNAL in the Golgi, enhancing their TL reactivity [23]. Analysis of ESAG6 expressed in *TbGT10⁻ᐟ⁻/TbGT8^{Flox/-}* and *TbGT10⁻ᐟ⁻/TbGT8⁻ᐟ⁻* mutant cells by Western blotting suggested that maturation of ESAG6 by modification of its *N*-glycans with pNAL is impaired (S2 Fig). To investigate this further, we treated affinity-purified TfR with endo-β-*N*-acetylglycosidase H (Endo H) which selectively removes oligomannose *N*-glycans from wild type, *TbGT10⁻ᐟ⁻/TbGT8^{Flox/-}* and *TbGT10⁻ᐟ⁻/TbGT8⁻ᐟ⁻* mutant TfR preparations. After treatment, the samples were subjected to SDS-PAGE in duplicate and analysed by TL blotting without (Fig 7, top panel, 1° TL blot) or with competing chitin hydrolysate (Fig 7, bottom panel, 1° TL/chitin

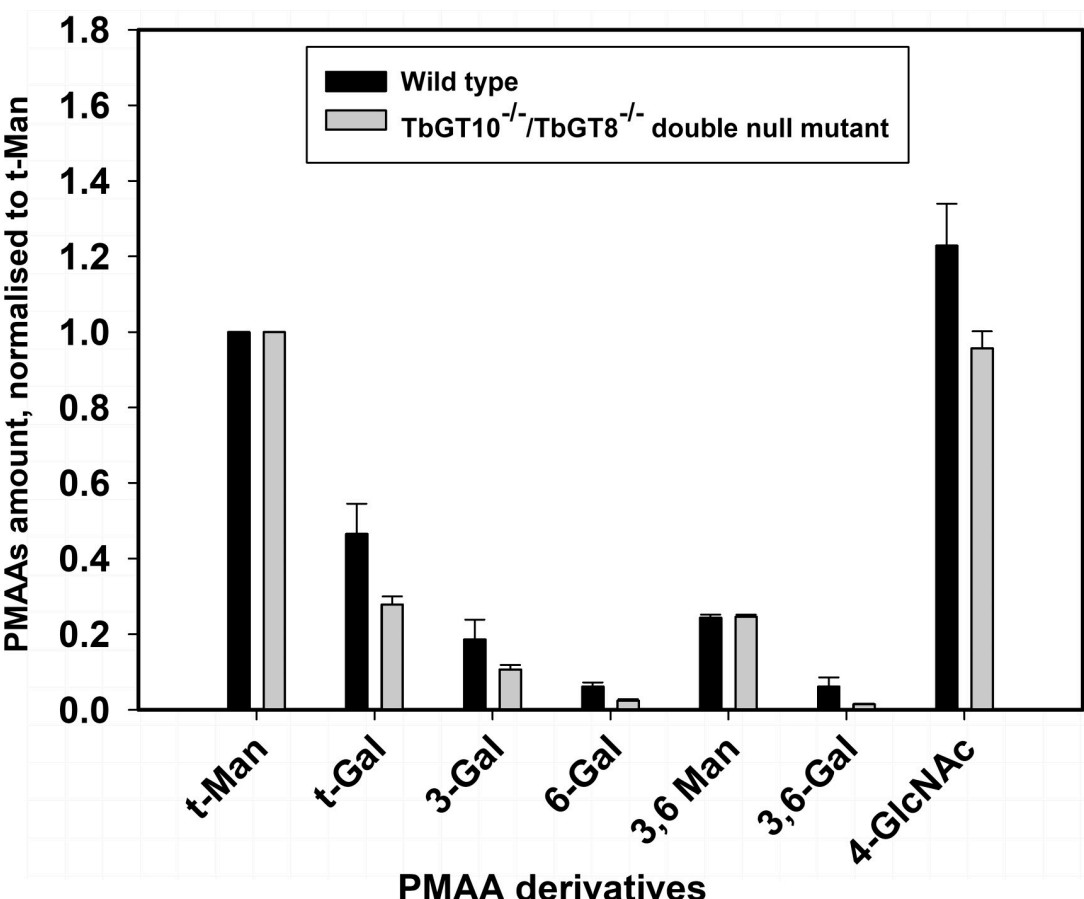

**Fig 6. Methylation linkage analysis of glycopeptides from BSF wild type (WT) and *TbGT8<sup>-/-</sup>/TbGT10<sup>-/-</sup>* double null (dKO) mutants confirm reduced linear pNAL and 3,6-GlcNAc branch synthesis.** Methylation linkage analysis of glycopeptides from BSF wild type (WT, black bars) and *TbGT10-/-/TbGT8-/-double null* (dKO, grey bars) mutants confirms reduced linear pNAL and 3,6-GlcNAc branch synthesis. The bar graph shows the percentage of partially methylated alditol acetate (PMAA) derivatives, compared to wild type PMAAs. All the PMAAs counts were first normalised to the derivative for non-reducing terminal-mannose (t-Man). The PMAAs were analysed using selected ion monitoring (SIM) on GC-MS, where characteristic PMAA fragment ions were used to collect and extract the data. (For WT, n = 6, 3 biological replicates with 2 technical replicates and for dKO, n = 4, 2 biological replicates with 2 technical replicates). The derivatives measured and the respective characteristic ions used for analysis are mentioned in Table 2.

blot) to confirm TL binding specificity. All primary blots were re-blotted with anti-TfR (2˚ αTfR blots, top and bottom panels) to ensure efficient pulldowns and Endo H digestion.

In wild type samples, both Endo H treated (partially deglycosylated ESAG6; dESAG6) and untreated ESAG6 showed strong reactivity in primary TL blots with a banding pattern and mobility that overlapped with the corresponding ESAG6 signals in anti-TfR secondary blots (Fig 7, compare lanes 1 and 3, 2 and 4; ESAG6 vs dESAG6). Endo H untreated ESAG7 was weakly reactive, while Endo H treated showed no reactivity (Fig 7, compare lanes 1 and 3, 2 and 4; ESAG7 vs dESAG7). The Endo H treated partially deglycosylated ESAG7 (dESAG7) (Fig 7, lane 2) signal does not overlap with the corresponding dESAG7 anti-TfR signal (Endo H+, lane 4). These results are fully in agreement with published work confirming that ESAG6 (which is strongly TL-reactive), unlike ESAG7 (which is weakly reactive), is modified by complex *N*-glycans containing linear pNAL [23]. In contrast, the absence of TbGT10 in the *TbGT10<sup>-/-</sup>/TbGT8<sup>Flox/-</sup>* mutant or both TbGT10 and TbGT8 in the *TbGT10<sup>-/-</sup>/TbGT8<sup>-/-</sup>* mutant resulted in loss of TL reactivity in ESAG6 (Fig 7, lanes 5, 6, 9, 10). The 2<sup>0</sup> aTfR blots (Fig 7,

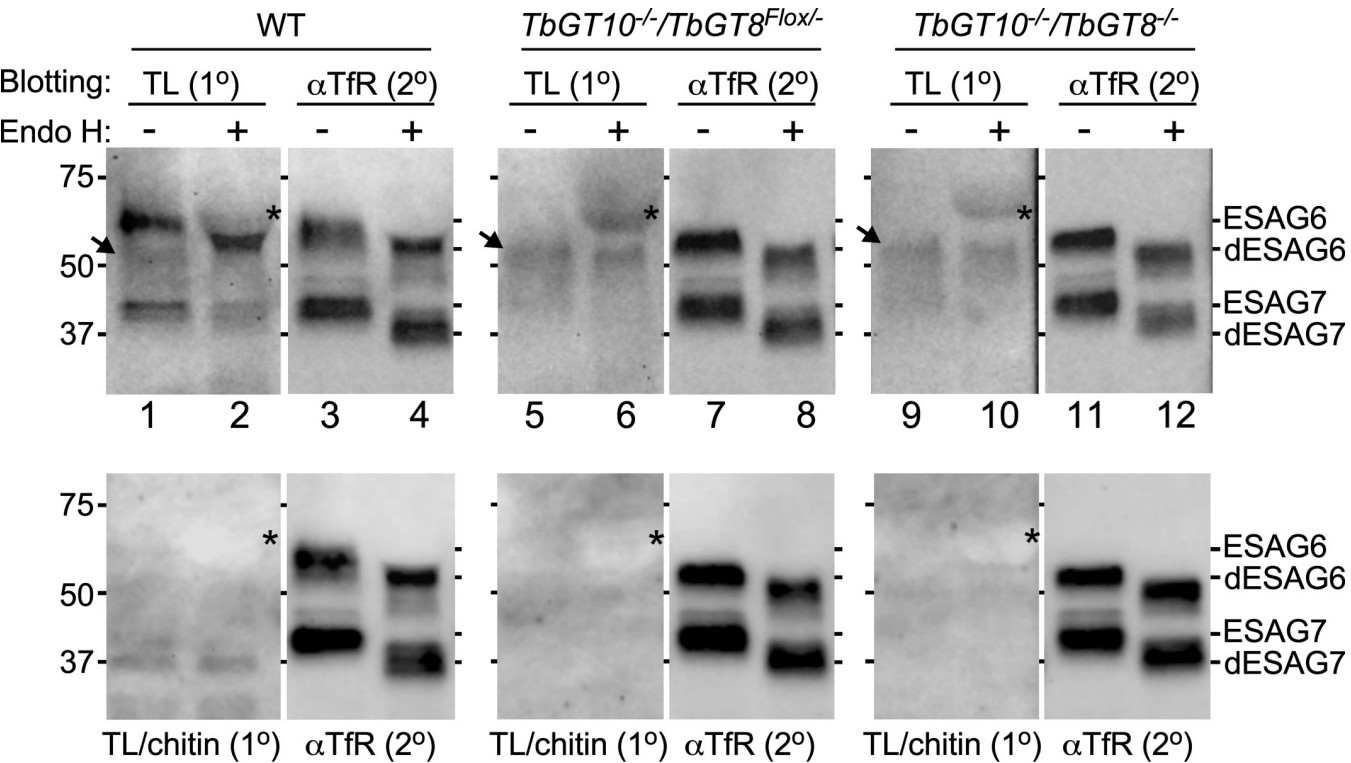

**Fig 7. Affinity purification of TfR, Endo H digestion, and TL blotting.** TfR was immunoprecipitated from whole cell lysates ($10^7$ cell equivalents/lane) of WT, *TbGT10^{-/-}/TbGT8^{Flox/-}* conditional null mutant, and *TbGT10^{-/-}/TbGT8^{-/-}* double null mutant cell lines. The affinity-purified precipitates were treated with (+) or without (-) Endo H to digest oligomannose glycoconjugates, and matched samples were subjected to blotting with TL::biotin only (top panel, TL $1^0$ blots) or TL::Biotin pre-incubated with competing chitin hydrolysate at 1:10 dilution (bottom panel, TL/chitin, $1^0$ blots). The primary blots were re-probed with anti-TfR (aTfR, $2^0$) without stripping to ensure efficient pulldowns and Endo H digestion. Mobilities of undigested ESAG6, ESAG7 (Endo H -) and Endo H de-glycosylated ESAG6, ESAG7 (dESAG6 and dESAG7, Endo H +) are shown on the right of each blot. Positions of molecular weight markers are shown on the left. Asterisks indicate the position of Endo H protein, while the arrowhead indicates mobility of a non-specific cross-reactive band. Data presented are representative of three independent biological replicates. Panels were digitally separated after image processing for clarity of presentation.

lanes 3,4,7,8,11,12) act as TfR loading and Endo H digestion controls. We attribute certain bands observed in the TL blots to non-specific cross-reacting bands (Fig 7, arrowheads), as these were also present in wild type samples. Importantly, the mobility of the non-specific bands did not overlap with the migration of ESAG6 in anti-TfR blots. Following TL blotting, non-specific detection of recombinant Endo H was also observed and is indicated by asterisks.

Collectively, these data indicate that the TfR ESAG6 subunit made in the absence of TbGT10 alone or of TbGT10 and TbGT8 together does not undergo extensive glyco-processing to yield TL-binding, pNAL-containing *N*-glycans. Moreover, in the case of ESAG6, no compensatory TbGTs appear to be able to restore any processing of ESAG6 to carry TL-binding pNAL-containing *N*-glycans.

The fact that ESAG6 from the *TbGT10^{-/-}/TbGT8^{Flox/-}* and *TbGT10^{-/-}/TbGT8^{-/-}* mutants failed to bind TL prompted us to investigate whether the absence of pNAL on TfR affects binding and uptake of its ligand. Lysates from wild and mutant cells were subjected to pulldowns with either anti-TfR antibodies or holo-Tf-coated beads. The pulldowns were resolved by SDS-PAGE, transferred to PVDF membranes, and immunoblotted with anti-TfR (Fig 8A). A number of conclusions can be made from these data: First, the anti-TfR blot revealed no obvious differences in the steady-state levels of the ESAG6 and ESAG7 TfR subunits (Fig 8A, compare lanes 1, 2, 3), suggesting that lack of pNAL does not affect TfR protein stability. Second,

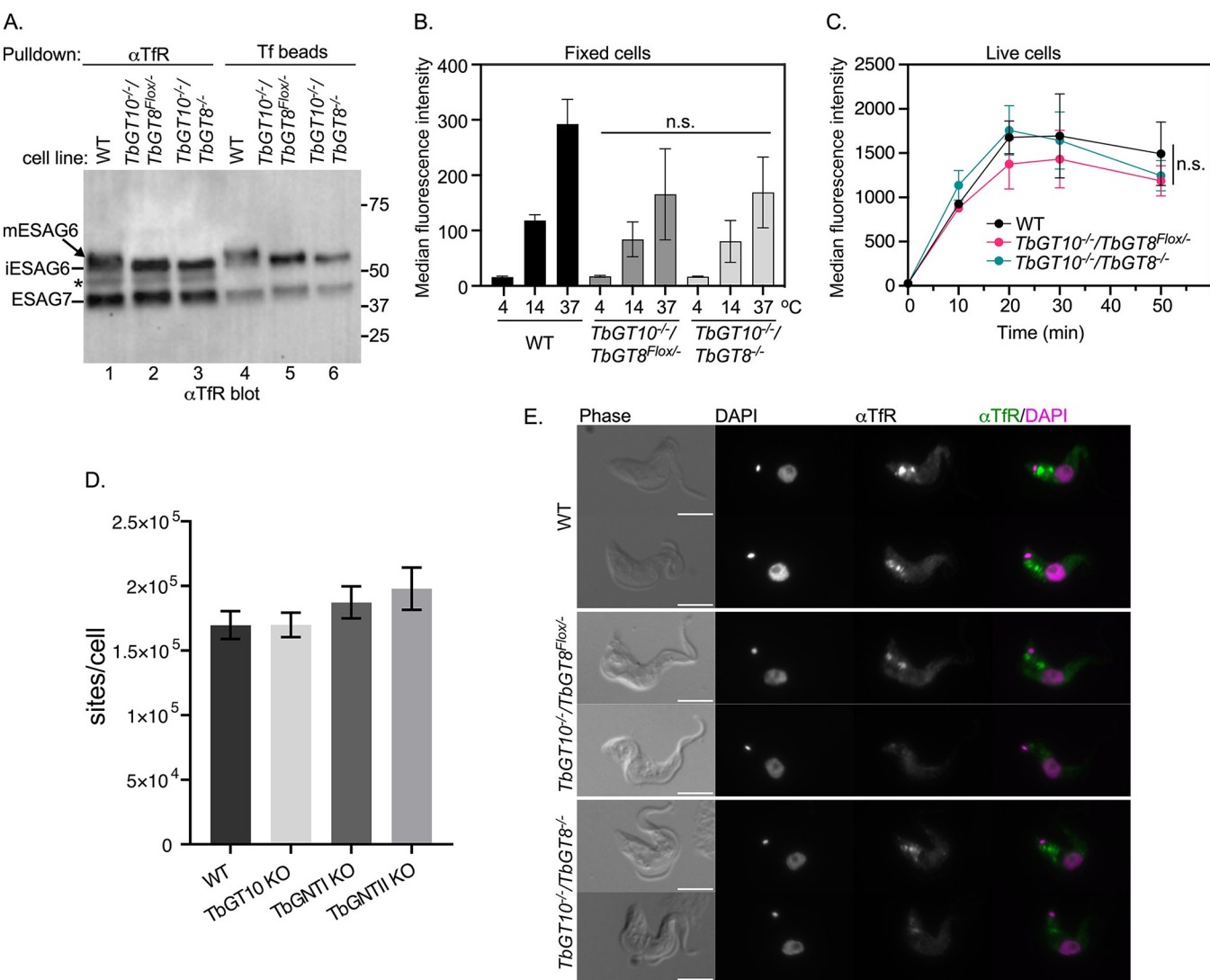

**Fig 8. Effect of TbGT knockout on TfR processing, endocytosis, and localisation. A.** Trypanosome cell extracts were prepared from WT, *TbGT10⁻/⁻/TbGT8^Flox/-* conditional null mutant and *TbGT10⁻/⁻/TbGT8⁻/⁻* double null mutant cell lines. $10^7$ cell equivalents per condition were affinity-precipitated with either anti-TfR (aTfR) or holo-transferrin (Tf beads), separated by SDS-PAGE, transferred to PVDF membranes, and blotted with aTfR antibodies. The mobilities of fully glycosylated mature ESAG6 (mESAG6), unprocessed immature ESAG6 (iESAG6), and ESAG7 are shown on the left. Molecular weight markers are shown on the right. **B.** Live cells from WT and KO mutants were incubated with Alexa488-conjugated holotransferrin (Tf::488) at 4, 14, and 37˚C for 30 min. Unbound Tf:488 was washed, the cells were formaldehyde fixed, and analysed by flow cytometry. Bar chart shows the median fluorescence intensity from 10,000 cells per condition. The difference in Tf::488 binding at 4˚C or uptake at 14 or 37˚C between WT and *TbGT10⁻/⁻/TbGT8^Flox/-* conditional null mutant or *TbGT10⁻/⁻/TbGT8⁻/⁻* double null mutant cells was not statistically significant (n.s.) as determined by unpaired t-tests. Error bars represent means ± SD, n = 3 biological replicates. **C.** To determine the kinetics of endocytosis, live cells were incubated with Tf::488 at indicated time intervals (x-axis), washed and analysed flow cytometry without fixing, to measure the rate of receptor-mediated endocytosis of Tf. Error bars represent means ± SD, n = 3 biological replicates. **D.** Binding of FITC-holotransferrin was measured in WT, *TbGT10* KO, *TbGT11* (TbGNTI) KO, and *TbGT15* (TbGNTII) KO mutants (sites/cell). Data are means ± SD (n = 3 biological replicates). **E.** Localisation of TfR in WT, *TbGT10⁻/⁻/TbGT8^Flox/-* conditional null mutant and *TbGT10⁻/⁻/TbGT8⁻/⁻* double null mutant cells. Microscopy was performed on formaldehyde fixed, permeabilised cells stained with DAPI (magenta) and anti-TfR (αTfR, green). Two representative cells are presented for each cell line. Arrowhead indicates flagellar pocket localisation of TfR. Scale-bar: 5 μM. Additional images of TfR localisation are presented in supplementary figure (S9 Fig).

only a fraction of total TfR associates with Tf-coated beads, consistent with the notion that some TfR was already saturated with Tf from the media prior to the pulldowns (Fig 8A, compare lanes 1–3 vs 4–6) [32]. Third, association to Tf beads confirmed the presence of functional heterodimers of ESAG6 and ESAG7 subunits, indicating that an absence of pNAL on *N-*

glycans does not affect TfR interaction with Tf *in vitro*. Fourth, the decrease in the size of ESAG6 in the mutants relative to wild type (Fig 8A, compare lane 1 vs 2 and 3; lane 4 vs 5 and 6) confirm that removal of TbGT10 alone or TbGT10 and TbGT8 together result in limited glyco-processing of ESAG6.

Next, we used flow cytometry to quantify uptake of fluorescent holotransferrin (Tf::488) in a large number of single cells *in vivo* (Fig 8B). The *TbGT10^{-/-}/TbGT8^{Flox/-}* and *TbGT10^{-/-}/TbGT8^{-/-}* mutant cells showed approximately 30% and 42% reductions in Tf uptake at 14°C and 37°C relative to wild type, respectively. However, these differences were not statistically significant. Given the variability of these experiments when cells were fixed prior to flow cytometry, we measured the actual rate of Tf:488 accumulation in live cells over 50 min and found no statistical difference in uptake between wild type and the *TbGT10^{-/-}/TbGT8^{Flox/-}* and *TbGT10^{-/-}/TbGT8^{-/-}* mutant cells (Fig 8C).

Our flow cytometry assay is not sensitive enough to measure Tf binding at 4°C [32], so we also assessed the number of binding sites using FITC:Tf. Here, we used wild type cells and *TbGT10^{-/-}*, *TbGT11^{-/-}* and *TbGT15^{-/-}* null mutant cells generated in previous studies [14,16]. *TbGT11* and *TbGT15* encode TbGnTI and TbGnTII, respectively, which add β1–2 linked GlcNAc residues to each arm of the $Man_3GlcNAc_2$ core, thus initiating complex *N*-glycan synthesis. Cells were incubated in the presence of FITC-conjugates, washed and cell extracts digested with Pronase to liberate FITC in order to measure total fluorescence [33] (Fig 8D). No statistically significant difference was detected in the binding of Tf between wild type and the three TbGT null mutants, indicating that altered pNAL synthesis in these cells has no detectable effect on cell surface Tf binding sites.

As an additional check for TfR function, the expression of RNA binding protein 5 (RBP5) was assessed in wild type cells and the *TbGT10^{-/-}/TbGT8^{Flox/-}* and *TbGT10^{-/-}/TbGT8^{-/-}* mutants (S8 Fig). RBP5 expression is upregulated upon iron starvation conditions, yet RBP5 levels were unchanged relative to wild type in our mutants, indicating that TfR endocytosis and iron scavenging is not impaired by pNAL loss [20,34,35].

Finally, to ensure that changes in the glyco-modifications of TfR did affect its cellular distribution, we performed immunofluorescent staining with anti-TfR antibodies. Microscopy revealed discrete foci between the central nucleus and the posterior kinetoplast, including a distinct signal in the flagellar pocket (Figs 8E, arrowheads and S9) [32]. This localisation is typical for functional TfR and appeared qualitatively similar in both mutants and WT, although there was a slight reduction in signal intensity in the TbGT mutants (S9 Fig). Despite the subtle differences, overall, these results indicate that TfR function is not impaired following loss of the TbGTs.

## High concentrations of N-acetylchito-oligosaccharides inhibit Tf receptor-mediated endocytosis

The hypothesis that TfR mediated endocytosis is pNAL-dependent is based principally on the reduced uptake of transferrin (Tf) in cells pre-treated with 15 mM of tri-*N*-acetyl-chitotriose and tetra-*N*-acetylchitotetraose [18]. This hypothesis contradicts our findings that reduced pNAL on TfR does not significantly affect endocytosis of Tf.

To assess this, we sought to reproduce the finding in [18] and to investigate concentration dependence. We performed our standard endocytosis assay to quantify the uptake of TL and Tf (receptor-mediated endocytic cargo) and dextran (fluid-phase cargo) in wild type cells using three dilutions of chitin hydrolysate (a mixture of *N*-acetylchito-oligosaccharides; $(\text{-4GlcNAcb1-})_n$) (Fig 9). We found that uptake of both TL and Tf (Fig 9A and 9B), but not dextran (Fig 9C), was inhibited at a 1:10 dilution of chitin hydrolysate. However, only TL

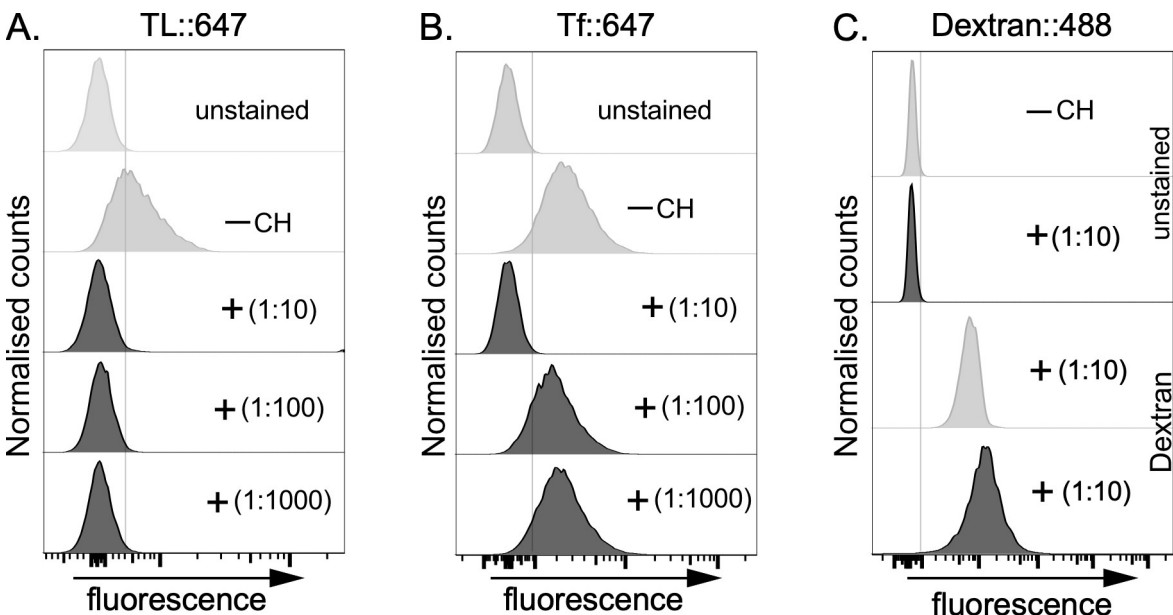

**Fig 9. Effect of chitin hydrolysate on TL, Tf and Dextran uptake.** Live, wild-type bloodstream form cells were incubated with (**A**) Alexa647-conjugated tomato lectin (TL::647), (**B**) Alexa647-conjugated transferrin (Tf::647), or with (**C**) Alexa488::Dextran at 37°C for 10 min to allow uptake. This was done in the absence (- CH, unstained, grey) or presence of varying dilutions (1:10 to 1:1000, black) of Chitin hydrolysate (+) as indicated in the plots (insets). The actual concentration of CH used is unknown, as this information is proprietary to the manufacturer of the product (Vector Laboratories, 2BScientific, UK). Histogram profiles show fluorescent intensities distributions of 10,000 cells per condition analysed by flow cytometry. Inhibition of TL and Tf uptake by chitin hydrolysate at a 1:10 dilution was consistent across three biological replicates in Figs 5 and 7, respectively. The Dextran experiment was performed once as control for fluid phase endocytosis.

uptake was inhibited at 1:100 and 1:1000 dilutions. These data reproduce the result in (18) that high concentrations of *N*-acetylchito-oligosaccharides inhibit Tf uptake but also show that lower concentrations of *N*-acetylchito-oligosaccharides do not, even when sufficient to block the binding of TL to cell surface (flagellar pocket) pNAL containing molecules.

## Discussion

Based on our previous investigations into trypanosomal GTs we have engineered a BSF *T. brucei* mutant deficient in pNAL synthesis by making a *TbGT10⁻/⁻/TbGT8^{Flox/-}* conditional null mutant where a floxed copy of *TbGT8* (*TbGT8^{Flox}*) can be excised in a *TbGT10⁻/⁻* null background upon exposure to rapamycin. Removal of both genes had little effect on cell viability and limiting dilution was applied to generate stable *TbGT10⁻/⁻/TbGT8⁻/⁻* double null mutant clones which were viable in cell culture and infectious to mice. Tomato lectin (TL) binds tightly to pNAL structures [30] and TL blots of cell lysates from the *TbGT10⁻/⁻/TbGT8^{Flox/-}* conditional null mutant grown under non-permissive (plus rapamycin), and from the *TbGT10⁻/⁻/TbGT8⁻/⁻* double null mutant, showed significantly reduced (but not abolished) TL binding. This was corroborated by methylation linkage analysis of Pronase glycopeptides from *TbGT10⁻/⁻/TbGT8⁻/⁻* cells, which showed a reduction (but not abolition) of 3,6-di-O-substituted-Gal, representative of -4GlcNAcβ1-6(-4GlcNAcβ1–3)Galβ1-branch points, of 4-O-substituted-GlcNAc and 6-O-substituted-Gal representative of -6Galβ1-4GlcNAcβ1-repeats, and 3-O-substituted-Gal representative of -3Galβ1-4GlcNAcβ1-repeats. These data suggest that other, as yet uncharacterised TbGTs [1], most likely those with the highest sequence similarities to TbGT10 and TbGT8, are also capable of UDP-GlcNAc: βGal β1–3 and UDP-GlcNAc: βGal β1–6 transferase activities.

Despite our inability to abolish pNAL synthesis entirely in the *TbGT10*[-/-]/*TbGT8*[-/-] double null cells, lectin blots for terminal β-Gal residues using RCA revealed that some glycoproteins were very substantially reduced in apparent molecular weight in the mutant. This included the essential cell surface-associated GPI anchored glycoprotein ESAG2 [36], known to normally bind to tomato lectin [18], and serine carboxypeptidase III (CBP1B) with likely lysosomal localisation based on analysis of a CPB1 homologue in *T. cruzi* [37]. Both ESAG2 and CBP1B are predicted to be heavily *N*-glycosylated (Table 1 and S4 Fig) and have apparent molecular weights of > 100 kDa in wild type cells [13,18], despite theoretical molecular weights of 56 and 52 kDa, respectively. Therefore, their detection at ~60 kDa in the *TbGT10*[-/-]/*TbGT8*[-/-] double null mutant is a likely consequence of the loss of all or most of their *N*-glycan pNAL modifications. A similar phenomenon was also observed for the endosomal/lysosomal protein p67 on BSF cells, where the molecular weight of the mature glycoprotein was reduced by 50 kDa in the *TbGT10*[-/-] null mutant [17,38]. Further, the data presented in this paper show that the ESAG6 subunit of the BSF TfR is significantly reduced in apparent molecular weight and no longer binds TL in the *TbGT10*[-/-]/*TbGT8*[Flox/-] and *TbGT10*[-/-]/*TbGT8*[-/-] mutants. A similar conclusion with respect to ESAG6 glycosylation and ability of TfR was also reported in [39]. Thus, all the evidence suggests that these proteins and others which are normally modified by polydisperse pNAL-containing complex *N*-glycans are still functional in the absence or depletion of pNAL structures.

The survival of the *TbGT10*[-/-]/*TbGT8*[-/-] double null mutant *in vitro* and *in vivo*, despite reductions in pNAL synthesis, appears contrary to the hypothesis of Nolan et al [18] which posits that receptor-mediated endocytosis of essential host molecules, like transferrin to supply iron, is pNAL-dependent. Instead, the main effect of deletion of TbGT8 and/or TbGT10 is the significantly impaired uptake of TL relative to WT but this is simply due to the reduction in TL binding sites.

The aforementioned pNAL-dependent endocytosis hypothesis was principally supported by the ability of 15mM *N*-acetylchitotriose and *N*-acetylchitotetraose to reduce the uptake of Tf by about 80%, and the uptake of low and high density lipoproteins (LDL and HDL) by about 50% [18]. We were able to reproduce the effects on Tf uptake using high concentrations of chitin hydrolysate (1:10 dilution), which impaired both TL and Tf uptake into cells. By contrast fluid phase uptake, measured with dextran, was unaffected at the highest concentration of chitin hydrolysate (1:10). These data recapitulate a specific effect on TfR mediated Tf endocytosis at high *N*-acetylchito-oligosaccharide concentrations. However, at lower chitin hydrolysate concentrations (1:100 and 1:1000 dilution) TL uptake was still robustly inhibited, but not that of Tf. This shows that under conditions where endogenous TL-like lectins (like exogenous TL) are likely to be fully ligated by chitin-oligomers, there is little or no effect on Tf uptake. Taken together with the evidence presented here for pNAL-independence of Tf binding an uptake by the parasite TfR, we recommend that other explanations for the effects of high-concentrations of chito-oligosaccharides on Tf uptake should be entertained and caution exercised in making pNAL-dependent endocytosis a default hypothesis.

Recently, *TbGT11*[-/-]/*TbGT15*[-/-] BSF *T. brucei* dKO mutants were successfully generated [40]. Despite the reduced elaboration of complex *N*-glycans in these mutants, they remain viable *in vitro*, exhibiting phenotypes similar to the dKO mutants from this study. Specifically, pNAL-modified glycoproteins, such as 115-kDa ectophosphatase (*Tb*AcP115), show a reduction in mass by ~50 kDa in mutants relative to WT, while still maintaining normal levels of fluid-phase endocytosis. Receptor-mediated endocytosis was also assessed in *TbGT11*[-/-]/*TbGT15*[-/-] dKO mutants using TbHpHbR, a GPI-anchored, pNAL modified receptor mediating uptake of the haptoglobin (Hp)–hemoglobin (Hb) complex and trypanolytic factor (TLF) at the FP. Unlike in our *TbGT10*[-/-]/*TbGT8*[-/-] dKO mutants, where TfR uptake is unaffected,

the uptake of HpHb is impaired in $TbGT11^{-/-}$/ $TbGT15^{-/-}$ dKO mutants, suggesting that efficient endocytosis of certain ligands is *N*-glycan dependent [40]. This discrepancy might be due to the absence of GT67 family compensatory activity in $TbGT11^{-/-}$/$TbGT15$-/- mutants, compared to the overall reduced but not complete loss of pNAL modifications in $TbGT10^{-/-}$/ $TbGT8^{-/-}$ mutants. Alternatively, the loss of complex *N*-glycan may impair TbHpHb ligand binding leading to reduced uptake. Thus, examining the effects on TbHpHb uptake in our mutant, as well as the rate of TfR endocytosis in $TbGT11^{-/-}$/ $TbGT15^{-/-}$ dKO mutant, will shed more light on the precise role of complex N-glycans in modulating ligand-receptor interactions in *T. brucei*.

Finally, although the deletion of TbGT8 and/or TbGT10 has provided useful insights into trypanosome cell biology, and questioned prior assumptions, the precise role(s) of the giant, complex pNAL-containing *N*-glycan modifications in *T. brucei* remain obscure.

## Methods

### Ethics statement

Animal studies were carried out under UK Home Office regulations (Project licence P4525BB4C) and the study plan approved by a Home Office Animals (Scientific Procedures) Inspector and approved by the University of Dundee ethics committee.

### Cultivation of trypanosomes

*Trypanosoma brucei brucei* Lister strain 427 bloodstream form parasites, expressing VSG variant 221 (MiTat1.2) and transformed to stably express T7 polymerase and the tetracycline repressor protein under G418 antibiotic selection, were used in this study. This genetic background will be referred to from hereon as wild-type (WT). Cells were cultivated in HMI-11 medium containing 2.5 µg/mL G418 at 37˚C in a 5% $CO_2$ incubator as described in [41].

### DNA Isolation and Manipulation

Plasmid DNA was purified from *Escherichia coli* DH5α competent cells (New England Biolabs) using a Qiagen Miniprep kit. Gel extraction and reaction clean-up was performed using Qiaquick kits (Qiagen). Custom oligonucleotides were obtained from Thermo Fisher. *T. brucei* genomic DNA was isolated from $\sim 2 \times 10^7$ bloodstream form cells using a DNeasy Blood & Tissue Kit (Qiagen) using standard methods.

*Generation of Gene Replacement Constructs-* A full list and descriptions of all primers (S1 Table) used in this study are available. The blasticidin deaminase (*BSD*) drug resistance cassette flanked by short regulatory elements from actin was generated by PCR-amplification using using Q5 DNA polymerase, a pNAT vector template and primers SMD402/3. ~500 bp 5' and 3' flanking regions of Tb427.10.12290 (NEB) with primers SMD400/1 and SMD404/5 respectively using (Lister strain 427, variant 221 strain) genomic DNA as a template. Primers were designed with 20 nucleotide overlap regions to enable Gibson Assembly into a linear, pUC19 vector amplified using primers SMD33/4. Amplicons were PCR purified (Qiagen) and 10 pmol of each used in a 4-fragment Gibson assembly reaction (NEB) to generate *TbGT8 5'_BSDr_TbGT8 3'_pUC19*. The first allele of *TbGT8* was replaced with the *BSD* drug resistance construct to generate a Δtb*gt8::BSD/TbGT8* single deletion mutant. To generate a *loxP*-flanked TbGT8 expression cassette Q5 polymerase PCR amplification of Tb427.10.12290 containing a 3' stop codon using oligonucleotides SMD355/6 to confer 5' *FseI* and 3' *BglIII* cloning sites for cloning into the *loxP* vector to generate *pSY45_pDS66 _TbGT8^Flox*. This construct was used as a template for PCR amplification using SMD351/2 whilst 5'-637 bp and 3' 588 bp

homologous flanks for Tb427.10.12290 were PCR amplified using SMD and SMD349/50 and SMD353/4, respectively. Each PCR amplicon contained 20 bp overlapping ends to facilitate Gibson assembly and generate *TbGT8 5'_pSY45_pDS66_TbGT8^Flox_TbGT8 3'_pUC19*. This construct was used to generate the Δtb*gt10::PAC/Δgt10/Δtbgt8::BSD/TbGT8^Flox* (*SSU diCre*) cell line hereon referred to as 'TbGT8^Flox' conditional knockout mutants.

## Transfection of bloodstream form T. brucei

Constructs for gene replacement and ectopic expression were purified, digested with appropriate restriction enzymes to linearize, precipitated, washed with 70% ethanol, and re-dissolved in sterile water. The linearized DNA was electroporated into *T. brucei* bloodstream form cells (Lister strain 427, variant 221) that were stably express T7 RNA polymerase and the tetracycline repressor protein under G418 selection. Cell transfection was carried out as described previously [41–43].

## Induction of diCre mediated gene deletion

Mid-log stage *TbGT10^-/-/TbGT8^Flox/-* conditional null mutant cultures (~1 x 10^6 cells/mL) were passaged to 2 x 10^3 cells/mL or 2 x 10^4 cells/mL and dosed with 100 nM rapamycin (Abcam) from a 1 mM stock solution in DMSO. Cells at late-log phase were harvested for analysis after 3 or 2 days respectively. Conditional gene deletion was assessed by PCR amplification of genomic DNA using Taq polymerase (NEB) and oligonucleotides SMD357/8 flanking the *TbGT8* locus. Hygromycin drug sensitivity was used as a proxy of *TbGT8^Flox* loss by seeding cells at a density of 2x 10^3 cells/mL and culturing for 3 days in the presence or absence of 100 nM rapamycin and/or hygromycin. Daily cell counting was performed to assess growth rates. A single clone (1.1) that gave a robust growth defect in the presence of hygromcyin was selected as the *TbGT10^-/-/TbGT8^Flox/-* conditional null mutant.

*Generation of double null mutants- TbGT10^-/-/TbGT8^Flox/-* conditional null mutant cells grown in the presence of 100 nM rapamycin for 3 days were serially diluted in 96 well plates in the presence of 100 nM rapamycin. Four clones were seeded at 2 x 10^3 cells/mL +/- hygromycin to confirm loss of *TbGT8_TK-HYG^Flox* array by drug sensitivity. *TbGT8^Flox* loss was confirmed by PCR analysis and a single clone (1.1) lacking both *TbGT10* and *TbGT8^Flox* (*TbGT10^-/-/TbGT8^-/-* double null mutant) was selected for further analysis.

## Mouse infectivity studies

Three groups of five female Balb/c mice each weighing between 18–25g were housed in standard holding cages with water and food available ad libitum throughout the study. Wild-type and *TbGT10^-/-/TbGT8^-/-* double null mutant bloodstream form trypanosomes were grown in HMI-11T media, washed in media without antibiotics and re-suspended at 1 x 10^6 cells/mL. 0.2 mL of the parasite suspension was injected intraperitoneally per animal. The ability of *TbGT10^-/-/TbGT8^-/-* double null mutant cells to establish infection in the blood relative to the WT control was assessed 1, 2 and 3 days post-infection by tail bleeding and cell counting using a Neubauer chamber in a phase contrast microscope.

## Western/ Lectin blotting

For Western and lectin blot analysis, 5 x 106–1 x 10^7 cells were lysed in lysis buffer (25mM Tris, pH 7.5, 100 mM NaCl, 1% Triton X-100) and solubilised in 1xSDS sample buffer containing 0.1 M DTT by heating at 55˚C for 20 min. Alternatively, cells were lysed by osmotic shock by incubating with water containing 0.1 mM TLCK, 1 µg/mL leupeptin and 1 µg/mL

aprotinin (pre-warmed to 37˚C) at 37˚C for 5 min. This releases all the cytosolic components and majority of VSG protein as soluble form VSG (sVSG). sVSG was liberated from cell ghosts by centrifugation at 12,000 g for 5 minutes and proteins extracted from the cell ghost pellet via lysis buffer and sample buffer extraction. Glycoproteins were resolved by SDS–PAGE (approx. $1 \times 10^7$ cell equivalents/lane) on NuPAGE *bis*-Tris 4–12% gradient acrylamide gels (Invitrogen) and transferred either to nitrocellulose or PVDF membranes (Invitrogen). Ponceau S staining confirmed equal loading and transfer. Glycoproteins were probed with 1.7 µg/mL biotin-conjugated RCA (RCA-120, Vector Laboratories, UK) in blocking buffer before or after pre-incubation with 30 mg/mL D-galactose and 30 mg/mL lactose to confirm specific RCA binding. Detection was performed using IRDye 680LT-conjugated Streptavidin and the LI-COR Odyssey Infrared Imaging System (LICOR Biosciences). For Western blotting, polyclonal antibodies raised against TfR and ISG65 were diluted 1:3000 and 1:2000, respectively in blocking buffer (50mM Tris-HCl pH7.4, 0.15M NaCl, 0.25% BSA, 0.05% (w/v) Tween-20, 0.05% $NaN_3$ and 2% (w/v) Fish Skin Gelatin). Detection with IRDye 800CW anti-Rabbit (LICOR Biosciences) was performed using 1:15,000 dilutions in blocking buffer.

For TL blotting, glycoproteins were probed for 1 hour using biotinylated-TL (1:5000 dilution, Vector Laboratories Inc.) in 3% BSA blocking buffer. To confirm the specificity of TL binding, biotinylated-TL was pre-incubated with a 1:10 dilution of chitin hydrolysate (Vector Laboratories Inc.). Membranes were washed three times with 1X PBST and then incubated with anti-biotin at a 1:5000 dilution for 1 hour at room temperature. Ponceau S staining served as loading control. Blots in Figs 5, 6, and 7 were developed using Super Signal ECL reagent (Thermo Fisher Scientific) and imaged with a ChemiDoc Gel Imaging MP System (Bio-Rad). Immunoblot analysis was carried out using Image Lab Software (Bio-Rad). Statistical analyses were performed by t-test in PRISM v10 (GraphPad Software, Inc., San Diego, CA). Differences were considered statistically significant at a *p*-value of <0.05.

## Analysis of FITC-Tf binding

Protocol from [33]. BSF *T. brucei* WT, TbGT10 KO, TbGT11 KO and TbGT15 KO cells were subbed at $1 \times 10^5$ cells /mL in 2 x 100 mL HMI-11 (10% FCS, no antibiotics) cultures and grown for 40 h prior to harvest. On the day of harvest 150 mL of each cell line was centrifuged at 1,000 g 10 mins at 4˚C and pellets re-suspended in 10 mL of ice-cold 1 X trypanosome dilution buffer (TDB; 5 mM KCl, 80 mM NaCl, 1 mM $MgSO_4$,20mM $Na_2HPO_4$, 2 mM $NaH_2PO_4$, 20 mM glucose, pH 7.4) in 15 mL falcon tubes. Cells were washed by centrifugation at 900 g for 10 mins at 4˚C and resuspension in a further 10 mL of ice-cold 1X TDB and centrifuged once more. All the supernatant was removed, and the pellet of cells resuspended in 1 mL ice cold TDB. Cells were diluted 1 in 100 by inoculating 10 µL of cell suspension in 990 uL 1x TDB and counted on a haemocytometer. Cells were adjusted to $2 \times 10^8$ cells/mL by addition of ice-cold 1 x TDB. A sample of this cell suspension was frozen for protein assays. Aliquots of 500 uL containing $10^8$ cells were transferred to 500 uL volumes containing 100 µg of FITC-holotransferrin (bovine). After incubating on ice for 15 minutes, cells were centrifuged at 11,700 x g for 30 sec at 4˚C. After removal of the supernatants, pellets were resuspended in 1 mL ice-cold TDB and centrifuged again at 11,700 x g for 30 sec at 4˚C. After discarding the supernatants, cells were resuspended in 1 mL ice-cold TDB and centrifuged again at 11,700 x g for 30 sec at 4˚C. Supernatants were then discarded, and the cell pellets frozen for proteolysis. FITC-holotransferrin standards for proteolysis were prepared by diluting 20 µL of the FITC-protein stock solution with 180 uL TDB to give final concentrations of 188.8 µg/mL and 183.3 µg/mL, respectively.

## Endocytosis assays

Endocytosis was performed as previously described [20]. Briefly, log-phase trypanosome cells were washed in Hepes-buffered saline (HBS; 50 mM Hepes KOH, pH 7.5, 50 mM NaCl, 5 mM KCl, 70 mM glucose) and pre-incubated ($10^6$/ml) in serum-free HMI9 with BSA (0.5 mg/ml) for 10 minutes at 37˚C. The cells were first incubated at 4˚C for 10 minutes to cool down, then 20 μg/ml TL DyLight-488 (Invitrogen) or 40 μg/ml hTf AF-488 (Invitrogen) was added, and cells were further incubated at 4˚C, 14˚C, or 37˚C for 10 minutes. After incubation, the cells were washed with cold 1x HBSG, fixed with 1% formaldehyde (SIGMA), and analysed by flow cytometry (10,000 cell counts/condition) using a BD LSRFortessa. Data were analysed using FlowJo (Tree Star). For uptake in live cells, 5 μg/ml TL TexasRed (Invitrogen) or 10 μg/ml hTf AF-647 (Jackson ImmunoResearch Laboratories, Inc.) was added and cells were incubated at 37˚C for 30 minutes. Flow cytometry analyses were identical to fixed cells.

## Affinity purification of TfR and Endo H treatment

TfR was affinity-purified using either anti-TfR antibodies or holo-transferrin-coated beads, following the methodologies described in [22,23]. A total of $1 \times 10^7$ cells were harvested and lysed in 1X RIPA buffer (25 mM Tris pH 7.4, 150 mM NaCl, 1% NP-40, 0.5% Na deoxycholate).1% SDS and 1 MM EDTA) containing the protease inhibitor cocktail described above. After clearing the lysates by centrifugation, the supernatant was mixed with either anti-TfR antibodies or a 50% slurry of Tf-beads and incubated overnight at 4˚C. After washing 3X with 1X RIPA buffer, the bound TfR was digested on beads with Endo H according to the manufacturer's recommendations (New England Biolabs). Eluates were analysed by lectin or Western blotting as described above.

## N-glycopeptide preparation for GC-MS analysis

To prepare *N*-glycopeptides of *T. brucei* bloodstream-form wild-type and *TbGT10⁻/⁻/TbGT8⁻/⁻* double null mutant cells, $5 \times 10^9$ cells were harvested and washed twice with 5 mM KCl, 80 mM NaCl, 1 mM $MgSO_4$, 20 mM $Na_2HPO_4$, 2 mM $NaH_2PO_4$, 20 mM glucose (pH 7.4) and depleted of sVSG using hypotonic lysis in 5 mL water (5 min, 37˚C). The sVSG depleted cell ghost pellets were recovered by centrifugation (12000 x g, 30 min, 4˚C) and resuspended in 1.5 mL of 20 mM ammonium bicarbonate and mixed with 50 μl of 10 mg/mL of freshly prepared Pronase (Calbiochem, #53702) dissolved in 5 mM calcium acetate and digested at 37˚C for 24 h. The digest was centrifuged at 12000 x g for 30 min to remove the cell ghost membranes and nuclei and the supernatant was incubated at 95˚C for 20 min to heat inactivate the Pronase and again centrifuged to remove particulates. The supernatant containing the Pronase digested glycopeptides was applied to a 30 kDa cut-off centrifugal filter (Amicon) and diafiltrated with water three times. The resulting aqueous filtrate was subjected to chloroform phase separation by mixing with equal volume of chloroform to remove any remaining lipid contaminants. The upper aqueous phase (Pronase glycopeptide fraction) was collected in a fresh tube and used for GC-MS methylation linkage analysis.

## Methylation linkage analysis

Samples were dried and subjected to permethylation using the sodium hydroxide method as described in [44]. The wild type (WT) glycans were permethylated with $^{13}CH_3I$, whereas the double null mutant glycans were permethylated with $CH_3I$. The permethylated glycans were then subjected to acid hydrolysis, $NaB[^2H]_4$ reduction, and acetylation to generate partially methylated alditol acetates (PMAAs) [44]. The PMAAs were analysed by GC-MS (Agilent

Technologies, 7890B Gas Chromatography system with 5977A MSD, equipped with Agilent HP5ms GC Column, 30 m X 0.25 mm, 0.25 μm). Selected ion monitoring (SIM) method was used to acquire the data using the characteristic fragment ions of PMAA derivatives. The most abundant and characteristic fragment ions with heavy (for WT) and light (dKO) isotopes were used to analyse the data. (t-Man and t-Gal: m/z 145/148; 3-Gal: m/z 161/163; 6-Gal: m/z 162/164; 3,6-Man and 3,6-Gal: m/z 234/236; 4-GlcNAc: m/z 159/160).

## RCA pulldown and protein identification of RCA reactive product

Soluble VSG fraction (sVSG) from $5 \times 10^7$ cell equivalents were collected and incubated with anti-VSG221 conjugated to Protein G Agarose (Pierce) for 1 hours at room temperature with rotation to deplete the sVSG of VSG221. Supernatants were removed from VSG221-beads by centrifugation and incubated with RCA-coupled Agarose (Vector Labs) overnight at 4°C with gentle rotation. The following day the RCA-agarose beads were washed three times in 1 x PBS and bound proteins eluted by heating in 1 x SDS sample buffer at 55°C for 30 minutes. Proteins were resolved by SDS-PAGE and either fixed using Quick Coomassie Stain (Neo Biotech) (approx. $2 \times 10^7$ cell equivalents/lane) or transferred to NTC membranes (approx. $1.4 \times 10^7$ cell equivalents/lane) and subjected to RCA lectin blotting before or after incubation with 30 mg/mL D-galactose and 30 mg/mL to localise the RCA reactive band. Slices corresponding to molecular weight of the RCA reactive band were cut from the Coomassie stained gel for LC–MS/MS protein identification at the FingerPrints Proteomics Facility (College of Life Sciences, University of Dundee).

## Supporting information

**S1 Fig. Genome sequencing validates knockout of both *TbGT8* and *TbGT10* in the *TbGT10*$^{-/-}$/*TbGT8*$^{-/-}$ double null mutant.** Genomic DNA harvested from WT, *TbGT10*$^{-/-}$/*TbGT8*$^{Flox/-}$ conditional null mutant (cKO) and *TbGT10*$^{-/-}$/*TbGT8*$^{-/-}$ double null mutant (dKO) cells was subjected to whole genome sequencing (30x coverage, paired end reads) and aligned to the *Trypanosoma brucei brucei* TREU927 reference genome. The plot shows mapped reads at the target genomic loci for *TbGT10* (Tb927.5.2760) and *TbGT8* (Tb927.10.12990).
(TIF)

**S2 Fig. The RCA reactive glycoprotein band expressed in *TbGT10*$^{-/-}$/*TbGT8*$^{-/-}$ double null mutants does not correspond to the transferrin receptor or invariant surface glycoprotein 65.** Cell ghosts were purified from soluble VSG (sVSG) by osmotic lysis. sVSG from WT, *TbGT8*$^{-/-}$null mutant, *TbGT10*$^{-/-}$/*TbGT8*$^{Flox/-}$ conditional null mutant and *TbGT10*$^{-/-}$/*TbGT8*$^{-/-}$ double null mutant cells were resolved by SDS-PAGE and transferred to nitrocellulose. Western blotting (green) with anti-transferrin receptor (TfR: upper panels) detecting both expression site associated genes (ESAG) 6 and 7 and anti-invariant surface glycoprotein 65 (ISG65: lower panels). Equal loading and transfer are demonstrated by Ponceau S staining (right). The region corresponding to the RCA reactive band is indicated (*) on the right and the molecular weight markers shown on the left. Products corresponding to mature (m)ESAG6, immature (i)ESAG6 and ESAG7 are indicated on the right. B. A schematic displaying the *N*-linked glycosylation states of immature and mature ESGA6 and ESAG7. Mature ESAG6 is reactive with tomato lectin (TL).
(TIF)

**S3 Fig. Protein identification of glycoproteins exhibiting increased RCA reactivity in *TbGT10*$^{-/-}$/*TbGT8*$^{-/-}$ double null mutants A.** Cell ghosts (pellet) were purified from VSG

fraction (supernatant) by osmotic lysis from WT and *TbGT10*$^{-/-}$/*TbGT8*$^{-/-}$ double null mutant cells. Pellet and supernatant fractions were resolved by SDS-PAGE and transferred to nitrocellulose in duplicate. Membranes were incubated with biotinylated RCA (red) without (RCA–Sugar: left) or with (RCA + Sugar: right) pre-incubation with 30 mg/ml galactose and lactose. Equal loading and sample transfer are demonstrated by Ponceau S staining. B. VSG (supernatant) fractions purified from WT, *TbGT10*$^{-/-}$/*TbGT8*$^{Flox/-}$ conditional null mutant and *TbGT10*$^{-/-}$/*TbGT8*$^{-/-}$ double null mutant mutants were depleted with anti-VSG221 conjugated agarose beads to remove VSG221 and non-bound material was incubated with RCA lectin agarose beads to purify RCA reactive glycoproteins. RCA agarose bound material was eluted by heating in SDS-sample buffer and resolved in triplicate lanes to enable RCA lectin blotting (upper panel) with RCA pre-incubated with 30 mg/mL galactose and lactose (RCA + Sugar) or not (RCA -Sugar) to enable Coomassie staining (lower panel). Gel slices from each sample corresponding to the RCA doublet were excised and submitted for proteomic analysis. The identified proteins are shown in Table 1.
(TIF)

**S4 Fig. *N*-linked glycan prediction analysis of ESAG2 and CBP1B.** The amino acid sequences from each protein were analysed using the N-glycosylation site prediction software [27]. Complex/paucimannose (red) and Oligomannose (green) glycosylation sites are indicated in the tables alongside their prediction scores and on the amino acid sequences. Signal peptides predicted by SignalP analysis are represented on the amino acid sequences in pink.
(TIF)

**S5 Fig. TbGT mutants are deficient in TL binding and uptake.** Binding and internalisation of TL were assessed by incubating cells with TL::Dylight488 (TL:488) at 4˚C for 5 minutes and then transferred to 37˚C for 10 minutes to activate endocytosis in the presence (+) or absence (-) of chitin hydrolysate, as described in Fig 5. Endocytosis was terminated on ice, and half of the cells were processed for flow cytometry (Fig 5B) while the other half were fixed with 2% paraformaldehyde, counterstained with DAPI for nuclei and kinetoplasts, mounted on slides, and imaged by fluorescence microscopy. The flagellar pocket is located adjacent to the kinetoplast, while endocytic compartments are situated between the central nucleus and the kinetoplast, indicated by large (nucleus) and small (kinetoplast) DAPI signals. Scale bar (black line): 5 μM.
(TIF)

**S6 Fig. GC-MS extracted ion chromatograms of wild type (upper) *TbGT8*$^{-/-}$ null mutant (middle) and *TbGT10*$^{-/-}$/*TbGT8*$^{-/-}$ double null mutant (lower) monosaccharide composition analysis.** A. The extracted *N*-glycopeptides were subjected to methanolysis and trimethylsilylation and the obtained methyl glycosides were analysed by GC-MS. 1 nmole of scyllo-inositol was used as an internal standard. B. Abundance of different monosaccharides plotted as a bar graph. Man: mannose, Gal: Galactose, GlcNAc: *N*-acetylglucosamine.
(TIF)

**S7 Fig. Consensus glycoconjugate structures in TbGT mutant *T. brucei*.** Representative schematic of the complex *N*-glycan structures synthesised in cells depleted of TbGT10, TbGT8 or both (double deletion), based on permethylation linkage analysis.
(TIF)

**S8 Fig. RBP5 expression is unchanged in *TbGT10*$^{-/-}$/*TbGT8*$^{Flox/-}$ conditional null mutants and *TbGT10*$^{-/-}$/*TbGT8*$^{-/-}$ double null mutants.** qRT-PCR analysis was performed to investigate *RBP5* and *TbGT8* transcript abundance in *TbGT10*$^{-/-}$/*TbGT8*$^{Flox/-}$ conditional null mutant

and *TbGT10*$^{-/-}$/*TbGT8*$^{-/-}$ double null mutant cells, relative to WT (dashed line). Data are means ± SD (n = 3 biological replicates with 3 technical replicates per n) ***P <0.0005 ****P <0.000005 as determined by unpaired T-test as compared with WT RNA expression levels. (TIF)

**S9 Fig. Localisation of TfR in WT,** *TbGT10*$^{-/-}$/*TbGT8*$^{Flox/-}$ **conditional null mutant and** *TbGT10*$^{-/-}$/*TbGT8*$^{-/-}$ **double null mutant cells.** Microscopy was performed on formaldehyde fixed, permeabilised cells stained with DAPI (magenta) and anti-TfR (αTfR, green). Representative cells are presented to show heterogeneity in TfR localisation between the cell lines. Arrowhead indicates flagellar pocket localisation of TfR. Scale bar = 5 μM. (TIF)

**S1 Table. List of oligonucleotide primers used in this study.** (TIF)

# Acknowledgments

We thank the resource unit and FingerPrints proteomic unit at the University of Dundee for providing assistance with the murine infection and protein identification, and Michele Tinti for analysing whole genome sequencing for validation of *TbGT8* and *TbGT10* knockout cell lines.

# Author Contributions

**Conceptualization:** Samuel M. Duncan, Rupa Nagar, Michael A. J. Ferguson, Calvin Tiengwe.

**Data curation:** Samuel M. Duncan, Carla Gilabert Carbajo, Rupa Nagar, Conor Breen, Michael A. J. Ferguson, Calvin Tiengwe.

**Formal analysis:** Samuel M. Duncan, Carla Gilabert Carbajo, Rupa Nagar, Qi Zhong, Conor Breen, Michael A. J. Ferguson, Calvin Tiengwe.

**Funding acquisition:** Michael A. J. Ferguson, Calvin Tiengwe.

**Investigation:** Samuel M. Duncan, Carla Gilabert Carbajo, Rupa Nagar, Qi Zhong, Conor Breen, Michael A. J. Ferguson, Calvin Tiengwe.

**Methodology:** Samuel M. Duncan, Carla Gilabert Carbajo, Rupa Nagar, Qi Zhong, Conor Breen, Michael A. J. Ferguson, Calvin Tiengwe.

**Project administration:** Michael A. J. Ferguson, Calvin Tiengwe.

**Resources:** Michael A. J. Ferguson, Calvin Tiengwe.

**Supervision:** Michael A. J. Ferguson, Calvin Tiengwe.

**Validation:** Samuel M. Duncan, Carla Gilabert Carbajo, Rupa Nagar, Qi Zhong, Michael A. J. Ferguson, Calvin Tiengwe.

**Visualization:** Samuel M. Duncan, Qi Zhong.

**Writing – original draft:** Samuel M. Duncan, Rupa Nagar, Michael A. J. Ferguson, Calvin Tiengwe.

**Writing – review & editing:** Samuel M. Duncan, Rupa Nagar, Michael A. J. Ferguson, Calvin Tiengwe.

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
