## [Decision Letter · Decision Letter 0]

26 May 2024

Dear Calvin,

Thank you very much for submitting your manuscript "Generation of a bloodstream form Trypanosoma brucei double glycosyltransferase null mutant competent in receptor-mediated endocytosis of transferrin" for consideration at PLOS Pathogens. As with all papers reviewed by the journal, your manuscript was reviewed by members of the editorial board and by several independent reviewers. The reviewers appreciated the attention to an important topic. Based on the reviews, we are likely to accept this manuscript for publication, providing that you modify the manuscript according to the review recommendations.

Sincerely,

Cynthia He

Academic Editor

PLOS Pathogens

Meera Nair

Section Editor

PLOS Pathogens

Michael Malim

Editor-in-Chief

PLOS Pathogens

orcid.org/0000-0002-7699-2064

Reviewer Comments (if any, and for reference):

Reviewer's Responses to Questions

**Part I - Summary**

Reviewer #1: In this work, Duncan and colleagues have investigated the function of two glycosyltransferases, TbGT10 and TbGT8, in the construction of long poly LacNAc units (pNAL) that are preferentially installed on flagellar pocket proteins in the bloodstream form of Trypanosoma brucei. In previous work, pNALs were proposed to function in receptor-mediated endocytosis of host proteins proteins such as transferrin. In this work, the authors have carefully removed either GT10 or GT10 and GT8 from BSFs and studied changes in glycosylation, infectivity, and TfR function. They show that deletion of GT10 alone or in combination with GT8 has no effect on BSF growth in vitro or infectivity. Dual or GT10 deletion alone removes most, but not all pNAL from cells, while the dual deleted cells have elevated RCA binding. Two proteins are identified as potential pNAL-modified proteins based off of changes in electrophoretic mobility in the GT10/TG8 dKO and LC-MS/MS. A more detailed analysis of TfR components ESAG6 and ESAG7 shows that the GT10 and dual KOs were deficient in pNAL assembly on ESAG6, which has recently been proposed to contain these oligosaccharides. The authors show that TfR lacking pNAL is still able to capture transferrin, suggesting that pNAL is not essential for receptor-mediated endocytosis as previously proposed. although they do confirm an effect on tomato lectin uptake in the presence of a competing oligosaccharide.

Overall, the work in the manuscript is of very high quality. The results in almost every case are carefully quantitated and rigorously controlled. The function of GT10 and GT8 in the production of pNAL is clearly demonstrated, which is an important result. The care taken with the experiments also shows that there is a small amount of residual pNAL that is likely produced by other GlyTs. The results show that pNAL is unlikely to have a direct role in receptor-mediated endocytosis, at least in the case of TfR.

Reviewer #2: This study investigates the function of two GT67 family glycosyltransferases and poly-N-acetyllactosamine (pNAL) protein glycosylation in T. brucei bloodstream forms. Previous studies have suggested that the addition of pNAC glycans to many flagellar pocket surface glycoproteins, including the transferrin receptor (TfR), is important for receptor internalization and parasite viability. To directly address this hypothesis, the authors have generated T. brucei knock-out lines lacking one or both pNAC generating glycosyltransferases, TbGT10 and/or TbGT8. Rigorous biochemical and phenotypic characterization of these mutants lines showed that these enzymes are indeed required for pNAL modification, although loss of pNAL modification in the double mutant was protein dependent and incomplete due to continued expression of other GT67 family members. Importantly, the authors were able to show that pNAL modification of the transferrin receptor, ESAG6, was largely inhibited in the knock-out lines, but that this was not associated with changes in receptor expression, trafficking and transferrin uptake. Overall, the study provides strong evidence that pNAL is not required for TfR function, contrary to the previous report. The analyses are very comprehensive, the paper well written and the conclusions strongly supported by the data. While the study leaves open the question what the function of pNAL is, it presents a significant advance in our understanding of the glycobiology of these parasites.

Reviewer #3: In this paper, the authors studied the synthesis and essentiality of poly-N-acetyllactosamine (pNAL) chains on complex N-glycans of glycoproteins from the bloodstream form of Trypanosoma brucei. For the synthesis of these pNAL chains, the authors previously identified and characterized glycosyltransferases TbGT10 and TbGT8 as responsible for most of the pNAL structures made by this parasite. In this work, the authors elegantly created a TbGT10 and TbGT8 double null mutant by generating a conditional null mutant with a single TbGT8 allele, which can be excised using the rapamycin-induced diCre system. Both the conditional null and double null mutant cells were shown to be viable in cell culture and infectious to mice.

Based on the activities of both TbGT10 and TbGT8, the authors propose that the loss of both enzymes would restrict the N-glycan variety to mannosylated (Man3-9GlcNAc2) and small bi-antennary complex N-glycans. Indeed, when comparing the binding of RCA lectin (which recognizes terminal beta-Gal residues) in standard blotting assays of lysates from WT, conditional null mutant, and double null mutant cells, they detected (i) a reduction in RCA binding in conditional null mutants grown in both permissive (- rapamycin) and restrictive (+ rapamycin) conditions compared to the WT, and (ii) the appearance of a band at ∼60 kDa when TbGT8 is removed in the presence of rapamycin. To identify the nature of these proteins by MS-proteomics, they performed RCA-agarose pulldown assays from hypotonically lysed cells pre-cleared of sVSG using anti-VSG IgG conjugated to protein G agarose. The bands were identified as GPI-anchored ESAG2 and CBP1B, a soluble lysosomal/endosomal serine protease. Both proteins contain identifiable N-glycosylation sequons.

While the overall glycobiology analysis of these mutants appears thorough and flawless, the most important contribution of this work, in my opinion, is that the double mutant allowed the authors to reinvestigate the essentiality of trypanosome pNAL repeats during the endocytosis of transferrin via the TfR at the flagellar pocket (Nolan and Pays). The authors used tomato lectin (TL), known to bind to flagellar pocket glycoproteins containing many pNAL chains. A TL blot showed reduced binding in the conditional null and double null mutants, corroborated by quantitative flow cytometry analysis of trypanosomes incubated at different temperatures with TL::488. Thus, after losing TbGT8, the expression of TL-reactive pNAL in double mutant cells appears drastically reduced, although not zero, suggesting that other uncharacterized trypanosomal GTs contribute to the formation of the residual pNAL repeats (demonstrated by MS analysis of pronase-generated glycopeptides). Furthermore, TL blots followed by re-probing with anti-TfR, pre/post treatment with EndoH, led the authors to conclude that ESAG6 made in the absence of TbGT10 or both TbGT10 and TbGT8 does not contain TL-binding, pNAL-containing N-glycans.

Importantly, no evidence of compensatory TbGTs adding pNAL chains to ESAG6 in mutant cells was found. The lack of pNAL repeats does not seem to affect the binding of transferrin, its internalization, or its cellular distribution (by IFA, which detected a reduction in signal intensity of TfR in the mutant cells). Moreover, endocytosis assays of TL and Tf in the presence of different concentrations of chitin hydrolysate (a competitive binder of TL) showed that the binding of Tf was only reduced at high concentrations of the inhibitor, but not at lower concentrations sufficient to block the binding of TL to pNAL chains.

Overall, this is a technically sound paper that makes a significant contribution to clarifying the role of pNAL repeats during the endocytosis of Tf in the bloodstream forms of T. brucei. I have only minor suggestions and comments.

**Part II – Major Issues: Key Experiments Required for Acceptance**

Reviewer #1: I have two related concerns that would improve the manuscript:

1. One of the most striking results is the specific inhibition of TL-488 in Figure 4B/C. It is presumed that the endocytosis is restored at the 14 and 37 C temperatures, but considering how striking the phenotype is I think it would be useful to confirm that the TL-488 is remaining confined to the flagellar pocket in the two mutants. Also, why did the authors choose to display median fluorescence intensity in 4C? In my experience, mean fluorescence of the population is the more commonly used measurement.

2. In Figure 8, I feel that the data in Figure D and E are somewhat contradictory. The number of Tf binding sites is not compromised, and perhaps seems somewhat elevated in knockouts. However, in E, assuming that equivalent lookup tables were used for the anti-TfR channel, it appears that there is a significant decline in TfR in the pocket region of the cells. From the data in 8A, it appears that there is not a decline in the total amount of TfR in the knockouts, so is it possible that the protein is not being effectively retained in the pocket? Otherwise, it seems that there is less TfR in the pocket of the mutants, but that it provides the same Tf binding capacity at in the WT? A more careful accounting of the TfR microscopy data may provide insight. Is there any TfR present on the cell surface in the KOs?

Reviewer #2: There are no significant points to address.

Reviewer #3: None

**Part III – Minor Issues: Editorial and Data Presentation Modifications**

Reviewer #1: I was a little surprised that ESAG6 is modified with pNAL but does not show as large a MW shift as the other pNAL-modified proteins that are discussed, which have 50-60 kDa attributable to the glycans. Is this due to fewer overall modified N-linked glycans or shorter pNAL repeats? Could this alter the function of the pNAL?

In Figure 1, I would perhaps consider leaving out the human pathway (A), as it does not directly pertain to the work presented.

Lines 198-203: Do the authors mean that the iterative deletions would lead to shorter glycans, or that the mutants could produce all the glycans they describe?

Lines 252-259: What criteria were used to identify these proteins from the RCA pulldown? Could they do some tagging or something to directly show change in glyTs affects the MW of the proteins?

Line 324: "EndoH", and "Endo H" are both used in the manuscript.

Line 324-326: This sentence is quite confusing, with the "unlike ESAG7" phrase being the main culprit. Consider revising, as it's very important for the interpretation of the rest of the figure.

Line 336-341: Perhaps a quick explanation of why some bands are protected from EndoH treatment could be useful here for non-glyco experts.

Fig 8E- I would include a scale bar for the micrographs.

Figure 8D-E. I think this data is contradictory. Unless the hypoglycosylated TfR is a significantly better binder of transferrin? Is there TfR on the cell surface, which could explain the decreased FP signal without seeing a loss in overall TfR signal on the WB?

Line 383: "wild type" not "wild"

Reviewer #2: (No Response)

Reviewer #3: Is there any transcriptomics or PCR evidence that in the absence of TbGT8 or TbGT10, or both, there is an increase in the expression of other parasite GTs?

Line 82: LLO abbreviation is repeated (line 75).

Line 215 and Fig. 4C: maybe I missed the point, but Fig 4C and Fig. 4B looks like the same experiment to me.

Line 259: please correct “endosmal”.

Line 290 and Fig. 5C: the authors claim by lectin blotting that the binding of TL is not affected in GT8 cells. By FC, however, they realize that the reduction is even greater in GT8 cells. Is there any statistical analysis backing this claim?

Line 336: it should be dESAG6, not dESAG.

Line 400: I think the authors refer to Fig. 8E, not to Fig. 8D.

Line 414 and Fig. 9: I found confusing the dextran explanation in the figure legend. Also, there’s no 1:1000 indication in the image, although it is mentioned in the text.

S2 Fig. has a very low-quality images, but this might be my own copy.

PLOS authors have the option to publish the peer review history of their article (what does this mean?). If published, this will include your full peer review and any attached files.

Reviewer #1: No

Reviewer #2: **Yes: **Malcolm McConville

Reviewer #3: No

Figure Files:

While revising your submission, please upload your figure files to the Preflight Analysis and Conversion Engine (PACE) digital diagnostic tool, https://pacev2.apexcovantage.com. PACE helps ensure that figures meet PLOS requirements. To use PACE, you mu

---

## [Editor Report · Decision Letter 1]

10 Jun 2024

Dear Clavin,

We are pleased to inform you that your manuscript 'Generation of a bloodstream form Trypanosoma brucei double glycosyltransferase null mutant competent in receptor-mediated endocytosis of transferrin' has been provisionally accepted for publication in PLOS Pathogens.

Best regards,

Cynthia Y He

Academic Editor

PLOS Pathogens

Meera Nair

Section Editor

PLOS Pathogens

Michael Malim

Editor-in-Chief

PLOS Pathogens

orcid.org/0000-0002-7699-2064
---

## [Editor Report · Acceptance letter]

24 Jun 2024

Dear Dr. Tiengwe,

We are delighted to inform you that your manuscript, "Generation of a bloodstream form Trypanosoma brucei double glycosyltransferase null mutant competent in receptor-mediated endocytosis of transferrin," has been formally accepted for publication in PLOS Pathogens.

Best regards,

Michael Malim

Editor-in-Chief

PLOS Pathogens

orcid.org/0000-0002-7699-2064